# TFGDA: Exploring Topology and Feature Alignment in Semi-supervised Graph Domain Adaptation through Robust Clustering

**Jun Dan[1*], Weiming Liu[1*], Chunfeng Xie[2], Hua Yu[3], Shunjie Dong[4], Yanchao Tan[5,6,7†]**
[1]Zhejiang University [2]Queen Mary University of London
[3]Dalian University of Technology [4]Shanghai Jiao Tong University [5]Fuzhou University
[6]Engineering Research Center of Big Data Intelligence, Ministry of Education
[7]Fujian Key Laboratory of Network Computing and Intelligent Information Processing
`danjun@zju.edu.cn, 21831010@zju.edu.cn, c.xie@qmul.ac.uk`
`yhiccd@mail.dlut.edu.cn, sjdong@sjtu.edu.cn, yctan@fzu.edu.cn`

## Abstract

Semi-supervised graph domain adaptation, as a branch of graph transfer learning, aims to annotate unlabeled target graph nodes by utilizing transferable knowledge learned from a label-scarce source graph. However, most existing studies primarily concentrate on aligning feature distributions directly to extract domain-invariant features, while ignoring the utilization of the intrinsic structure information in graphs. Inspired by the significance of data structure information in enhancing models' generalization performance, this paper aims to investigate how to leverage the structure information to assist graph transfer learning. To this end, we propose an innovative framework called TFGDA. Specially, TFGDA employs a structure alignment strategy named STSA to encode graphs' topological structure information into the latent space, greatly facilitating the learning of transferable features. To achieve a stable alignment of feature distributions, we also introduce a SDA strategy to mitigate domain discrepancy on the sphere. Moreover, to address the overfitting issue caused by label scarcity, a simple but effective RNC strategy is devised to guide the discriminative clustering of unlabeled nodes. Experiments on various benchmarks demonstrate the superiority of TFGDA over SOTA methods.

## 1 Introduction

With the rise of deep learning, node classification techniques have made significant progress in diverse fields. However, due to the distribution shift issue, well-trained models often suffer severe performance degradation when applied directly to a new domain. Graph transfer learning (GTL) [1] has been proposed to tackle this issue by transferring domain-invariant features from a labeled source graph to an unlabeled target graph, effectively boosting the model's performance on the target graph.

Although current studies on GTL have made significant strides, they often rely on the assumption that all nodes in the source graph are labeled. However, this ideal assumption does not hold true in many scenarios, as annotating the entire source graph is time-consuming, especially for large-scale networks. Therefore, in this paper, we focus on a more realistic application scenario known as semi-supervised graph domain adaptation (SGDA) [2], where the source graph only has a few labeled nodes. Utilizing the transferable knowledge acquired from the label-scarce source graph to enhance the model's adaptation performance on the target graph is the most crucial challenge for SGDA.

---

[*] Equal Contribution, [†] Corresponding Author.

38th Conference on Neural Information Processing Systems (NeurIPS 2024).

However, most existing studies [3, 4, 1] tend to focus on directly aligning feature distributions across domains to extract domain-invariant node features, while overlooking the utilization of the intrinsic structure information in graphs. Notably, the recent advancements in unsupervised learning [5, 6, 7, 8, 9] have showcased the significance of data structure information in enhancing models' generalization. Considering the complex topological structure information presented in the graph, this paper seek to leverage these critical structure information to assist the transfer of shared node knowledge. To this end, we propose an innovative SGDA framework named **TFGDA** that employs a Subgraph Topological Structure Alignment (**STSA**) strategy to encode structure information into latent space. Specially, STSA utilizes persistent homology (PH) [10] to extract the topological structure information of the input and latent spaces, and then align the topological structures of two space, significantly improving the model's transfer performance. Furthermore, current works [2, 11, 4] primarily utilize adversarial training to reduce domain discrepancy. However, adversarial training is an unstable process that may destroy the discriminative details hidden in features [12], thereby affecting the transfer of shared knowledge. To remedy this issue, we propose a Sphere-guided Domain Alignment (**SDA**) strategy, aiming to achieve more stable domain alignment. Concretely, SDA initially maps node features to the spherical space. Then, geodesic projection [13] is utilized to project spherical features onto multiple great circles, where the spherical sliced-Wasserstein (SSW) distance [14] is employed to quantify the feature distributions discrepancy across domains.

More importantly, in the SGDA scenarios, due to the label scarcity of source graph, well-trained model on only a few labeled source nodes is likely to encounter overfitting. Consequently, it may make ambiguous or even incorrect predictions for certain target graph nodes located near the decision boundaries or far from their corresponding class centers. To address this overfitting issue, we devise a Robustness-guided Node Clustering (**RNC**) strategy to effectively enhance the model's robustness. RNC aims to guide the discriminative clustering of unlabeled nodes by maximizing the mutual information between the soft cluster assignment of the original node and its perturbed version, significantly improving the model's generalization performance on the target graph.

In summary, the main contributions are listed as follows:

**(1)** To the best of our knowledge, this is first attempt to utilize the intrinsic topological structure information hidden in graphs to assist GTL. A novel STSA strategy is proposed to preserve the topological structure information in latent space.

**(2)** A strategy named SDA is introduced to stably align the node feature distributions across domains.

**(3)** To address the overfitting issue, a simple but effective RNC strategy is devised to guide the discriminative clustering of unlabeled nodes.

**(4)** Experimental results show that our TFGDA outperforms SOTA methods on various benchmarks.

## 2 Related Works

**Graph Transfer Learning.** GTL [15, 16] has gained widespread attention for relieving the burden of collecting labeled data for new tasks. Early studies usually use source nodes to pre-train expressive models for related tasks in target domain [17, 18, 19, 20, 21]. To enhance model's generalization, recent studies have shifted their emphasis to domain adaptation [11, 22, 23]. There are two main ways to extract domain-invariant node features: (1) Using adversarial training to enforce domain confusion [16, 4, 2, 24]; (2) Minimizing the statistical distance between two domains [25, 26, 27, 28, 29]. However, existing works tend to focus on domain alignment while overlooking the utilization of structure information in graphs.

**Semi-supervised Learning on Graphs.** Semi-supervised learning on graphs addresses node classification with a small fraction of labeled nodes. Previous works [30, 31, 32] commonly adopt the message passing paradigm to extract discriminative features. Recent studies have explored various techniques, including adversarial training [33, 34], data augmentation [35, 36], continuous graph [37], contrastive learning [27] and meta learning [38, 39] to further enhance model's generalization.

**Persistent Homology (PH).** PH is an essential method in topology for extracting structure information from point clouds [40, 41]. Recently, PH has shown significant advantages in various areas, including signal processing[42], shape matching[43], and design of network [44]. Some studies have investigated its differentiability [45, 46]. Recent works have also explored its potential in image segmentation [47, 48], action/image recognition [49, 50, 51, 52, 53], and evaluation of GANs [54].

# 3 Preliminaries: Persistent Homology (PH)

PH is a method used to capture the topological structure of complex point clouds as a scale parameter $\rho$ is varied. In this section, we briefly introduce some key concepts. Further details on PH can be found in [10, 40].

**Notation.** $\mathcal{X} := \{x_i\}_{i=1}^m$ denotes a point cloud and $\omega$ is a distance metric over $\mathcal{X}$.

**Vietoris-Rips (VR) Complex.** The VR complex [55] is a unique simplicial complex constructed from a set of points, providing an approximation of the underlying space's topology. The VR complex of $\mathcal{X}$ at scale $\rho$, denoted as $\mathcal{V}_\rho(\mathcal{X})$, contains all simplices of $\mathcal{X}$. Each component of $\mathcal{X}$ satisfies the distance constraint: $\omega(x_i, x_j) \leq \rho$ for any $i, j$. Additionally, the VR complex exhibits a nesting property: $\mathcal{V}_{\rho_i} \subseteq \mathcal{V}_{\rho_j}$ for any $\rho_i \leq \rho_j$, which enables us to track the evolution progress of simplical complex as $\rho$ increases.

**Persistence Diagram (PD).** The PD $dgm$ is a multi-set of points $(g_1, g_2)$ in the Cartesian plane $\mathbb{R}^2$, encoding lifespan information of topological features. Concretely, it summarizes the birth time $g_1$ and death time $g_2$ information of each topological feature with a homology group. The birth time $g_1$ indicates the scale of feature creation and death time $g_2$ refers to the scale of feature destruction.

# 4 Methodology

## 4.1 Problem Definition

**Source Domain Graph**: The source graph is defined as $\mathcal{G}^s = (\mathcal{V}^{s,l}, \mathcal{V}^{s,u}, Y^{s,l}, A^s, X^s)$, where $\mathcal{V}^{s,l}$ is the labeled node set, and $\mathcal{V}^{s,u}$ is the remaining unlabeled node set in $\mathcal{G}_s$. $Y^{s,l} \in \mathbb{R}^{|\mathcal{V}^{s,l}| \times K}$ denotes the label matrix of $\mathcal{V}^{s,l}$, where $K$ is the number of node classes. If a node $n_i^s \in \mathcal{V}^{s,l}$ belongs to the $k$-th class, $y_{i,k}^s = 1$; otherwise, $y_{i,k}^s = 0$. $A^s \in \mathbb{R}^{N^s \times N^s}$ is an adjacency matrix, where $N^s = |\mathcal{V}^{s,l}| + |\mathcal{V}^{s,u}|$ is the number of nodes in $\mathcal{G}^s$. If there is an edge between nodes $n_i$ and $n_j$, the value of $A_{ij}^s$ is set to 1; otherwise, it is set to 0. $X^s \in \mathbb{R}^{N^s \times e}$ indicates an attribute matrix, where $e$ is the dimension of node attributes. Notably, $|\mathcal{V}^{s,l}|$ is much smaller than $|\mathcal{V}^{s,u}|$ in the SGDA setting.

**Target Domain Graph**: Similarly, the target graph is represented as $\mathcal{G}^t = (\mathcal{V}^t, A^t, X^t)$, which is a completely unlabeled graph with an unlabeled node set $\mathcal{V}^t$. $A^t \in \mathbb{R}^{N^t \times N^t}$ is an adjacency matrix, and $X^t \in \mathbb{R}^{N^t \times e}$ is a node attribute matrix, where $N^t = |\mathcal{V}^t|$ denotes the number of nodes in $\mathcal{G}_t$.

**Semi-Supervised Graph Domain Adaptation (SGDA)**: Given a partially labeled source graph $\mathcal{G}^s$ and an unlabeled target graph $\mathcal{G}^t$, the goal of SGDA is to precisely annotate target graph nodes by utilizing transferable knowledge learned from the limited labeled source nodes [2, 16].

## 4.2 Network Architecture

The architecture of our TFGDA model is depicted in Figure 1. It consists of two components: a graph convolutional network (GCN)-based feature extractor $\mathcal{F}$ and a node classifier $\mathcal{C}$. Mathematically, given an input graph $\mathcal{G} = (\mathcal{V}, A, X)$, the node features extracted by $\mathcal{F}$ is denoted as $H = \mathcal{F}(\mathcal{G}) \in \mathbb{R}^{|\mathcal{V}| \times d}$, and it is further normalized to map onto a spherical space $\mathbb{S}_r^{d-1}$ to obtain spherical features $Z \in \mathbb{R}^{|\mathcal{V}| \times d}$, where $d$ is the feature dimension, $r$ is the radius and $|\mathcal{V}|$ denotes the number of nodes in $\mathcal{G}$. The classification probability predicted by $\mathcal{C}$ is denoted as $\Psi = \mathcal{C}(Z) \in \mathbb{R}^{|\mathcal{V}| \times K}$.

To capture more precise adjacency relationships of graph $\mathcal{G}$, we compute the positive point-wise mutual information (PPIM) between nodes following [2]. Specially, for a given graph $\mathcal{G} = (\mathcal{V}, A, X)$, we utilize random walk to sample a collection of paths on $A$ and generate a frequency matrix $R$. Based on $R$, we can compute the PPIM matrix $\mathbb{P}$ as follows:

$$\mathbb{P}_{ij} = \frac{R_{ij}}{\sum_{i,j} R_{ij}}, \; \mathbb{P}_{i,*} = \frac{\sum_j R_{ij}}{\sum_{i,j} R_{ij}}, \; \mathbb{P}_{*,j} = \frac{\sum_i R_{ij}}{\sum_{i,j} R_{ij}},$$
$$P_{ij} = \max\{\log(\frac{\mathbb{P}_{ij}}{\mathbb{P}_{i,*} \times \mathbb{P}_{*,j}}), 0\}, \tag{1}$$

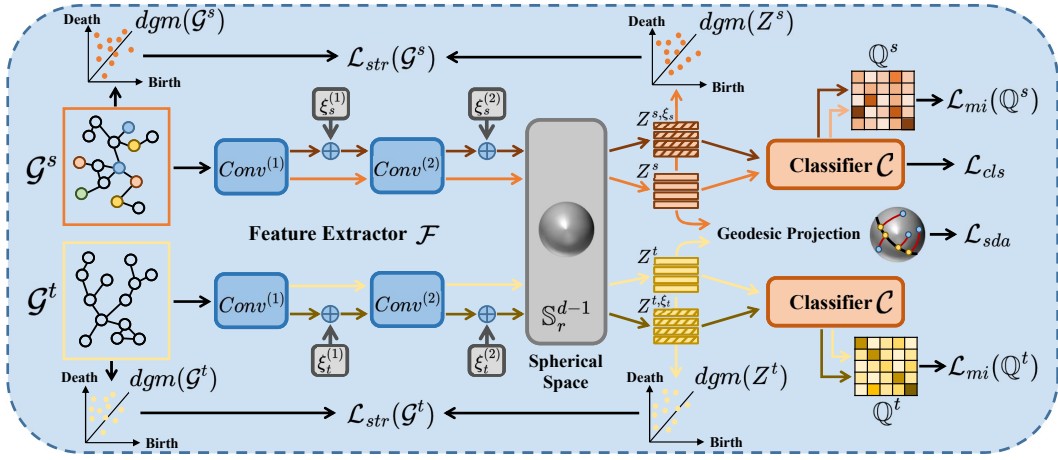

Figure 1: Global overview of the TFGDA model. STSA strategy encodes critical structure information of graphs into spherical space ($\mathcal{L}_{stsa} = \mathcal{L}_{str}(\mathcal{G}^s) + \mathcal{L}_{str}(\mathcal{G}^t)$), greatly improving the model's generalization. SDA strategy aims to extract domain-invariant node features by minimizing domain discrepancy on sphere ($\mathcal{L}_{sda}$). Moreover, to effectively solve the overfitting issue, RNC strategy is introduced to guide the discriminative clustering of unlabeled nodes ($\mathcal{L}_{rnc} = \mathcal{L}_{mi}(\mathbb{Q}^s) + \mathcal{L}_{mi}(\mathbb{Q}^t)$).

where $P_{ij}$ denotes the positive mutual information between nodes $n_i$ and $n_j$, which quantifies the topological proximity between nodes. A higher value of $P_{ij}$ indicates a strong connection between $n_i$ and $n_j$. Then, the output of the $l$-th GCN layer $Conv^{(l)}(\cdot)$ is defined as:

$$H^{(l)} = Conv^{(l)}(P, H^{(l-1)}) = \sigma(D^{-\frac{1}{2}}\widetilde{P}D^{-\frac{1}{2}}H^{(l-1)}W^{(l)}), \qquad (2)$$

where $D$ is the diagonal degree matrix of $P$, and $\widetilde{P} = P + I$ ($I$ is a identity matrix). $W^{(l)}$ refer to the trainable parameters of the $l$-th layer, $\sigma(\cdot)$ is an activation function, and $H^{(0)} = X$. Thus, the feature extractor $\mathcal{F}$ can be constructed by sequentially stacking $L$ layers of GCN $Conv^{(l)}(l = 1, 2, \cdots, L)$. Given the source labeled node set $\mathcal{V}^{s,l}$, the classification loss on the source graph $\mathcal{G}^s$ is defined as:

$$\mathcal{L}_{cls} = \frac{1}{|\mathcal{V}^{s,l}|} \sum_{n_i^s \in \mathcal{V}^{s,l}} \mathcal{L}_{ce}(\mathcal{C}(\mathcal{F}(n_i^s)), y_i^s). \qquad (3)$$

where $\mathcal{L}_{ce}$ represents the standard cross-entropy loss.

### 4.3 Subgraph Topological Structure Alignment

As mentioned in Section 1, the graph consists of numerous nodes (e.g., *ACMv9* has over 9000 nodes), thereby containing rich structure information. Inspired by the significance of data structure information in enhancing models' generalization [56, 51, 6], we seek to leverage such critical structure information to facilitate the learning of domain-invariant features.

To achieve this goal, we treat the graph as a point cloud and attempt to directly capture its underlying topological structure using PH. However, we inevitably encounter huge computational burden due to the complex attributes and adjacency relationships of graph. Fortunately, Refs. [57, 58] have indicated that the properties of graph can be well preserved in its multiple local subgraphs. To this end, we propose a Subgraph Topological Structure Alignment (**STSA**) strategy to encode structure information of input space into the latent space. For a given graph $\mathcal{G} = (\mathcal{V}, A, X)$, STSA first sample $a$ subgraphs $\left\{\hat{\mathcal{G}}_1, \hat{\mathcal{G}}_2, \cdots, \hat{\mathcal{G}}_a\right\}$ using random walk, and then employ PH to capture the intrinsic topological structure information of each subgraph $\hat{\mathcal{G}}_i$.

Specially, for each subgraph $\hat{\mathcal{G}}_i$, we represent its spherical features extracted by $\mathcal{F}$ as $\hat{Z}_i \in \mathbb{R}^{|\hat{\mathcal{V}}_i| \times d}$, which are obtained by indexing the local nodes features from the complete nodes features $Z \in \mathbb{R}^{|\mathcal{V}| \times d}$. Here $|\hat{\mathcal{V}}_i|$ denotes the number of nodes in each subgraph. Then we construct the VR complexes

$\mathcal{V}_\rho(\hat{\mathcal{G}}_i)$ and $\mathcal{V}_\rho(\hat{Z}_i)$ for point clouds $\hat{\mathcal{G}}_i$ and $\hat{Z}_i$, employ PH to extract their topological structures, and obtain their corresponding PDs $dgm(\hat{\mathcal{G}}_i)$ and $dgm(\hat{Z}_i)$ respectively. To align the topological structures of the input and latent spherical spaces, we adopt the 1-Wasserstein distance $\mathcal{W}_1$ to measure the discrepancy between two PDs (i.e., $dgm(\hat{\mathcal{G}}_i)$ for input space and $dgm(\hat{Z}_i)$ for spherical space), which aims to seek the optimal transport plan $\gamma^*$ between two PDs:

$$
\begin{aligned}
\gamma^* &= \arg\min_\gamma \mathcal{W}_1(dgm(\hat{\mathcal{G}}_i), dgm(\hat{Z}_i)) \\
&= \arg\min_\gamma \sum_{(\alpha,\beta)\in\gamma} \|\alpha - \beta\|_\infty
\end{aligned}
\tag{4}
$$

where $\alpha \in dgm(\hat{\mathcal{G}}_i)$, $\beta \in dgm(\hat{Z}_i)$, and $\|\cdot\|_\infty$ is the $l_\infty$ distance. After obtaining $\gamma^*$, the local structure discrepancy $\mathcal{L}_{str}^{sub}(\hat{\mathcal{G}}_i)$ between two spaces of $\hat{\mathcal{G}}_i$ can be calculated as:

$$
\mathcal{L}_{str}^{sub}(\hat{\mathcal{G}}_i) = \sum_{(\alpha,\beta)\in\gamma^*} \|\alpha - \beta\|_2^2.
\tag{5}
$$

Hence, we can estimate the global structure discrepancy between the input and spherical spaces of graph $\mathcal{G}$ by aggregating the local structure discrepancy of all subgraphs:

$$
\mathcal{L}_{str}(\mathcal{G}) = \sum_i \mathcal{L}_{str}^{sub}(\hat{\mathcal{G}}_i)
\tag{6}
$$

In the SGDA scenario, the STSA strategy will be applied to both the $\mathcal{G}^s$ and $\mathcal{G}^t$, and its loss function is defined as:

$$
\mathcal{L}_{stsa} = \mathcal{L}_{str}(\mathcal{G}^s) + \mathcal{L}_{str}(\mathcal{G}^t).
\tag{7}
$$

Notably, preserving topological structure information has the potential to guide unlabeled nodes towards achieving discriminative clustering, thereby promoting the learning of transferable node features, as verified in Section 5.3.

Although many GCNs-based methods have already been proposed to exploit graph structure information to promote the learning of features, these methods are not effective in addressing the SGDA task. Specifically, these GCNs-based methods [59, 60, 61, 62] typically mine the graph structure information in the deep feature space by designing well-crafted GCN architectures or introducing some complex modules. However, recent studies [27, 63, 64, 16] have pointed out that GCNs are insufficient in capturing the sophisticated structure information in graph, which means that the graph structure information may be lost or destroyed after passing through the GCNs-based feature extractors. Thus, directly mining graph structure information from the deep feature space is a suboptimal way, which affects the learning of transferable node features in our SGDA setting.

The proposed STSA strategy aims to extract the graph structure information directly from the input space and encode these powerful information into the latent spherical space by aligning the topological structures of the two spaces. This method does not lose or destroy the graph structure information during training. Furthermore, our STSA strategy does not introduce any changes to the network architecture, effectively avoiding an increase in model's complexity and ensuring its adaptability to integration with other methods.

### 4.4 Sphere-guided Domain Alignment

As mentioned in Section 1, adversarial training has been widely adopted by existing GTL models to reduce domain discrepancy. However, it is an unstable process that may destroy the discriminative information hidden in node features, thereby impacting the learning of shared features.

To tackle this issue, we propose a Sphere-guided Domain Alignment (SDA) strategy that achieves stable alignment of cross-domain node features distributions in the spherical space. Our SDA strategy mainly comprises three steps: **(1)** Map node features onto the sphere space $\mathbb{S}_r^{d-1}$. **(2)** Use geodesic projection [13] to project node features from $\mathbb{S}_r^{d-1}$ to multiple great circles. **(3)** Compute the feature distributions discrepancy on great circles and minimize it during training.

**Step (1):** Motivated by the effectiveness of spherical features in improving model's transfer performance [65], we first normalize node features $H \in \mathbb{R}^{|\mathcal{V}|\times d}$ extracted by $\mathcal{F}$ with $z_i = r\frac{h_i}{\|h_i\|}$ to obtain

the spherical features $Z \in \mathbb{R}^{|\mathcal{V}| \times d}$ in the sphere space $\mathbb{S}_r^{d-1} = \left\{ z_i \in \mathbb{R}^d : \|z_i\|_2 = r \right\}$, where $h_i$ and $z_i$ is the $i$-th row of $H$ and $Z$, respectively. Notably, Ref. [66] has proved that a proper radius $r$ is lower bounded by parameters $\tau$ and $K$:

$$r \geq \frac{K-1}{K} \ln \frac{(K-1)\tau}{1-\tau} \tag{8}$$

where $\tau$ denotes the expected minimal classification probability of class center and $K$ is the number of classes. In this work, $\tau$ is set to 0.999, and radius $r$ is set to the lower bound.

**Step (2):** Previous studies [67, 68, 69, 70] have shown the superiority of optimal transport in aligning feature distributions. Let $\Omega$ be a probability space and $\mu$, $\nu$ be two probability measures in $\mathcal{P}(\Omega)$. For any $q \geq 1$, the $q$-Wasserstein distance between $\mu$ and $\nu$ is defined as:

$$\mathcal{W}_q^q(\mu, \nu) = \inf_{\gamma \in \Pi(\mu, \nu)} \int_{\Omega \times \Omega} \mathcal{M}^q(z, g) d\gamma(z, g) \tag{9}$$

where $\Pi(\mu, \nu) = \{\gamma \in \mathcal{P}(\Omega \times \Omega) | \pi_{1\#}\gamma = \mu, \pi_{2\#}\gamma = \nu\}$ is the set of couplings, $\pi_1$ and $\pi_2$ denote the two marginal projections of $\Omega \times \Omega$ to $\Omega$, $\#$ denotes the push-forward operator, and $\mathcal{M} : \Omega \times \Omega \to \mathbb{R}^+$ is a geodesic metric. However, we find that directly using the classical Wasserstein distance to compute features distributions discrepancy on the sphere $\mathbb{S}_r^{d-1}$ is computationally expensive, due to numerous nodes in the graph.

For more efficient calculations, we utilize geodesic projection $I^U$ to project node features lying on $\mathbb{S}_r^{d-1}$ to $b$ great circles $\{\mathbb{C}_1, \mathbb{C}_2, \cdots, \mathbb{C}_b\}$. On the hypersphere, great circles [71] are circles whose diameter is equal to that of the sphere, and they correspond to the geodesics. Specially, the geodesic projection $I^U$ is determined by $U$:

$$I^U(z) = U^\top \arg\min_{g \in \mathrm{span}(UU^\top) \cap \mathbb{S}_r^{d-1}} \mathcal{M}_{\mathbb{S}_r^{d-1}}(z, g) = \arg\min_{\varrho \in \mathbb{S}_r^1} \mathcal{M}_{\mathbb{S}_r^{d-1}}(z, U\varrho), \forall U \in \mathbb{V}_{d,2}, \forall z \in \mathbb{S}_r^{d-1}. \tag{10}$$

where $\mathcal{M}_{\mathbb{S}_r^{d-1}}(z, g) = \arccos(\langle z, g \rangle)$, and $\mathbb{V}_{d,2} = \left\{ U \in \mathbb{R}^{d \times d}, U^\top U = I_2 \right\}$ is the Stiefel manifold [72].

**Step (3):** Next, we utilize the spherical sliced-Wasserstein (SSW) distance [14] to measure the feature distributions discrepancy of two domain on multiple great circles, which can be formulated as:

$$SSW_q^q(u, v) = \int_{\mathbb{V}_{d,2}} \mathcal{W}_q^q(I_\#^U \mu, I_\#^U \nu) d\sigma(\mathrm{U}), \tag{11}$$

where $\sigma$ is the uniform distribution over $\mathbb{V}_{d,2}$. In SSW, the geodesic metric $\mathcal{M}^q(z, g)$ in Wasserstein distance $\mathcal{W}_q^q(\mu, \nu)$ is defined as the geodesic distance [71] $\mathcal{M}_{\mathbb{S}_r^1}(z, g) = \min(|z - g|, r - |z - g|)$.

In practice, it's common to approximate the source and target feature distributions using samples $(z_i^s)_{i=1}^{N^s}$ and $(z_j^t)_{j=1}^{N^t}$ (i.e., through the empirical approximations $\tilde{\mu} = \frac{1}{N^s} \sum_{i=1}^{N^s} \delta_{z_i^s}$ and $\tilde{\nu} = \frac{1}{N^t} \sum_{j=1}^{N^t} \delta_{z_j^t}$), where $\delta$ is the Dirac function. As a result, the node features distributions discrepancy between two domains can be measured as:

$$\mathcal{L}_{sda} = SSW_p^p(\tilde{\mu}, \tilde{\nu}) \approx \frac{1}{b} \sum_{m=1}^b \mathcal{W}_q^q(\tilde{\mu}, \tilde{\nu}) \tag{12}$$

where $b$ is the number of projections. As training progresses, SDA strategy gradually reduces domain discrepancy, making the learning of domain-invariant features easier.

## 4.5 Robustness-guided Node Clustering

Due to the label scarcity in $\mathcal{G}^s$, the model is prone to overfitting when solely relying on $\mathcal{L}_{cls}$ for optimization, severely degrading the model's generalization performance on $\mathcal{G}^t$. To alleviate this overfitting issue, we devise a novel Robustness-guided Node Clustering (**RNC**) strategy to enhance the model's robustness by guiding the discriminative clustering of unlabeled nodes. Specially, RNC first introduces trainable shift parameters $\xi_s = \left\{ \xi_s^{(1)}, \xi_s^{(2)}, \cdots, \xi_s^{(L)} \right\}$ and $\xi_t = \left\{ \xi_t^{(1)}, \xi_t^{(2)}, \cdots, \xi_t^{(L)} \right\}$ to perturbs the source and target node features respectively, at each layer of $\mathcal{F}$:

$$H^{s/t, \xi_{s/t}, (l)} = \begin{cases} Conv^{(l)}(P^{s/t}, X^{s/t}) + \xi_{s/t}^{(l)}, & l = 1 \\ Conv^{(l)}(P^{s/t}, H^{s/t, (l-1)}) + \xi_{s/t}^{(l)}, & 1 < l \leq L \end{cases} \tag{13}$$

where $H^{s,\xi_s,(l)}$ and $H^{t,\xi_t,(l)}$ denotes the perturbed source and target node features encoded by $Conv^{(l)}$ respectively, and each $\xi_{s/t}^{(i)}$ is specific to the output of $Conv^{(l)}$. The shift parameters $\xi_s$ and $\xi_t$ are defined as randomly initialized multi-layer parameter matrices. After perturbation, based on $H^{s,\xi_s,(L)}$ and $H^{t,\xi_t,(L)}$, we can obtain the perturbed source node spherical features $Z^{s,\xi_s}$ and the perturbed target node spherical features $Z^{t,\xi_t}$ respectively.

For brevity, we will omit domain-specific notations in the following text. Ideally, regardless of how the node feature is perturbed, the model's prediction for it should remain unchanged because its class label has not changed. To achieve this goal, RNC aims to maximize the mutual information between the soft cluster assignment (i.e., classification prediction of classifier $\mathcal{C}$) of the spherical features $Z$ and its perturbed version $Z^\xi$, capturing their intrinsic invariant information between $Z$ and $Z^\xi$.

Concretely, given a node spherical feature $z_i$, the classification probability predicted by $\mathcal{C}$ is denoted as $\Psi(z_i) \in \mathbb{R}^K$, that can be viewed as the distribution of a discrete random variable $\phi$ over $K$ classes: $\Gamma(\phi = k|z) = \Psi_k(z_i)$. Let $\phi$ and $\phi^\xi$ denote the cluster assignment variables of $z_i$ and $z_i^\xi$, respectively. Then, their conditional joint distribution is defined as: $\Gamma(\phi = k, \phi^\xi = k^\xi|z_i, z_i^\xi) = \Psi_k(z_i) \cdot \Psi_{k^\xi}(z_i^\xi)$. After marginalization, the joint probability distribution can be formulated as a matrix $\mathbb{Q} \in \mathbb{R}^{K \times K}$:

$$\mathbb{Q} = \frac{1}{|\mathcal{V}|} \sum_{i=1}^{|\mathcal{V}|} \Psi(z_i) \cdot \Psi(z_i^\xi)^\top, \tag{14}$$

where $\mathbb{Q}_{kk^\xi} = \Gamma(\phi = k, \phi^\xi = k^\xi)$, $\mathbb{Q}_k = \Gamma(\phi = k)$, and $\mathbb{Q}_{k^\xi} = \Gamma(\phi^\xi = k^\xi)$. To preserve the equivalence between pairs $(z_i, z_i^\xi)$ and $(z_i^\xi, z_i)$, matrix $\mathbb{Q}$ is typically symmetrized using $(\mathbb{Q} + \mathbb{Q}^\top)/2$. In this way, the mutual information [73, 74] between the soft cluster assignment of $Z$ and $Z^\xi$ can be computed as:

$$\mathcal{L}_{mi}(\mathbb{Q}) = \sum_{k=1}^{K} \sum_{k^\xi=1}^{K} \mathbb{Q}_{kk^\xi} \cdot \ln \frac{\mathbb{Q}_{kk^\xi}}{\mathbb{Q}_k \cdot \mathbb{Q}_{k^\xi}}, s.t., ||\xi^{(l)}||_F \le \varpi, \forall \xi^{(l)} \in \xi. \tag{15}$$

where $\varpi$ is a coefficient that controls the scale of feature perturbation.

In the SGDA scenario, all target domain nodes $\mathcal{V}_t$ and source domain nodes $\mathcal{V}^{s,l} \cup \mathcal{V}^{s,u}$ are used to calculate matrices $\mathbb{Q}^t$ and $\mathbb{Q}^s$, respectively. Therefore, the objective function $\mathcal{L}_{rnc}$ of RNC can be expressed as:

$$\mathcal{L}_{rnc} = \mathcal{L}_{mi}(\mathbb{Q}^s) + \mathcal{L}_{mi}(\mathbb{Q}^t),$$
$$s.t., ||\xi^{s,(l)}||_F \le \varpi, \forall \xi^{s,(l)} \in \xi^s, ||\xi^{t,(l)}||_F \le \varpi, \forall \xi^{t,(l)} \in \xi^t. \tag{16}$$

Notably, we leverage source labeled nodes $\mathcal{V}^{s,l}$ in RNC as they can significantly guide the discriminative clustering of unlabeled nodes in the right direction. With the help of RNC, more and more intrinsic invariant features of nodes are extracted, which greatly promotes the learning of transferable features. Moreover, unlike previous studies that employ pseudo-labels strategy [2] or conditional entropy term [75] to guide the learning of unlabeled nodes, our RNC strategy does not involve any pseudo-labels and naturally avoids degenerate clustering solutions (see Figure 3 for further analysis).

### 4.6 Model Optimization

In summary, the total objective of TFGDA can be expressed as follows:

$$\min_{\mathcal{F}, \mathcal{C}, \xi^s, \xi^t} \mathcal{L}_{cls} + \eta \mathcal{L}_{sda} + \varepsilon \mathcal{L}_{stsa} - \lambda \mathcal{L}_{rnc}$$
$$s.t., ||\xi^{s,(l)}||_F \le \varpi, \forall \xi^{s,(l)} \in \xi^s, ||\xi^{t,(l)}||_F \le \varpi, \forall \xi^{t,(l)} \in \xi^t. \tag{17}$$

where hyper-parameters $\eta$, $\varepsilon$ and $\lambda$ are used to balance the contributions of the corresponding term.

### 4.7 Theoretical Analysis

The theoretical analysis of our method is based on the theory of domain adaptation (DA) [76, 77].

Formally, let $\mathcal{H}$ be the hypothesis space. Given two domains $\mathcal{S}$ and $\mathcal{T}$, the probabilistic bound of error of hypothesis $h$ on the target domain is defined as: $\psi_\mathcal{T}(h) \le \psi_\mathcal{S}(h) + \frac{1}{2}d_{\mathcal{H}\Delta\mathcal{H}}(\mathcal{S}, \mathcal{T}) + \mu^*$, where the expected error on the target domain $\psi_\mathcal{T}(h)$ are bounded by three terms: **(1)** the expected

error on source domain $\psi_{\mathcal{S}}(h)$; **(2)** the $\mathcal{H}\Delta\mathcal{H}$-divergence between the source and target domains $d_{\mathcal{H}\Delta\mathcal{H}}(\mathcal{S}, \mathcal{T})$; **(3)** the combined error of ideal joint hypothesis $\mu^* = \min_{h' \in \mathcal{H}} \psi_{\mathcal{S}}(h') + \psi_{\mathcal{T}}(h')$.

The goal of DA is to lower the upper bound of the expected target domain error $\psi_{\mathcal{T}}(h)$. Note that in unsupervised domain adaptation (UDA), minimizing $\psi_{\mathcal{S}}(h)$ can be easily achieved with source label information, as source domain samples are completely annotated. However, in our SGDA setting, due to the label scarcity of source domain, the model prone to overfitting when solely relying on the source domain classification loss $\mathcal{L}_{cls}$ for optimization. Therefore, we introduce the **RNC** strategy ($\mathcal{L}_{rnc}$) to address the overfitting issue, with the aim of guiding $\psi_{\mathcal{S}}(h)$ towards further minimization.

Most DA methods mainly focus on reducing the domain discrepancy $d_{\mathcal{H}\Delta\mathcal{H}}(\mathcal{S}, \mathcal{T})$, such as utilizing techniques like adversarial learning, MMD, optimal transport [78, 79, 80, 81], and CORAL. In comparison to these methods, our **SDA** strategy ($\mathcal{L}_{sda}$) effectively eliminates the feature norm discrepancy in spherical space $\mathbb{S}_r^{d-1}$ and guide a more stable alignment of feature distributions. Furthermore, considering that graph data contains rich structure information that encodes complex relationships among nodes and edges, and existing GTL methods usually adopt GCNs-based feature extractors to learn domain-invariant node features. However, recent studies [27, 63, 64, 16] have pointed out that GCNs are insufficient in capturing the sophisticated structure information in graph, which seriously affects the transfer of domain-invariant knowledge and consequently limits the model's generalization ability. To solve this problem, we thus propose the **STSA** strategy ($\mathcal{L}_{stsa}$) to align the topological structures of the input space and the spherical space, in order to facilitate the GCNs-based feature extractors to capture more domain-invariant node features. Consequently, the combination of **SDA** ($\mathcal{L}_{sda}$) and **STSA** ($\mathcal{L}_{stsa}$) strategies further promotes the minimization of the domain discrepancy $d_{\mathcal{H}\Delta\mathcal{H}}(\mathcal{S}, \mathcal{T})$.

Notably, $\mu^*$ is expected to be extremely small, and therefore it is often neglected by previous methods. However, it is possible that $\mu^*$ tends to be large when the cross-domain category distributions are not well aligned [82, 83]. In this paper, we leverage the **RNC** strategy ($\mathcal{L}_{rnc}$) to guide both labeled nodes and unlabeled nodes toward achieving robust clustering, effectively promoting the fine-grained alignment of category distributions and ensuring that $\mu^*$ remains at a relatively small value.

In summary, our proposed method not only minimizes the source expected error $\psi_{\mathcal{S}}(h)$ and domain discrepancy $d_{\mathcal{H}\Delta\mathcal{H}}(\mathcal{S}, \mathcal{T})$, but also keeps $\mu^*$ at a small value, thereby ensuring a low upper bound.

## 5 Experiments

### 5.1 Setup

**Datasets.** Our experiments involve three real-world graphs: *ACMv9* (**A**), *Citationv1* (**C**), and *DBLPv7* (**D**), obtained from ArnetMiner [84]. Since these graphs have varying sets of node attributes, we union their attribute sets and adjust the attribute dimension to 6775 following [2]. Each node is assigned a five-class label, determined by its relevant research areas. Six typical transfer tasks are considered in our experiments: **A→C**, **A→D**, **C→A**, **C→D**, **D→A** and **D→C**. Due to the page size limitation, more settings and implementation details are placed on **Appendix**.

**Compared Methods.** We compare TFGDA with several SOTA **(1)** graph semi-supervised learning methods and **(2)** graph domain adaptation methods as Ref.[2]: **(1)** **GCN** [30], **GSAGE** [31], **GAT**[32], **GIN** [85], **(2)** **DANN** [86], **CDAN** [12], **UDA-GCN** [75], **AdaGCN** [1], **CoCo** [27], **StruRW** [22] and **SGDA** [2]. DANN$_{GCN}$ and CDAN$_{GCN}$ are two variants that adopt GCN-based feature extractor.

### 5.2 Results and Discussion

Following [2], to showcase the superiority of our TFGDA, we report its performance on the challenging scenario, where only **5%** of the nodes in the source graph are labeled. **Micro-F1** and **Macro-F1** are employed as evaluation metrics, and the classification results on the target graph are gathered in Table 1. For all transfer tasks, we run each experiment 5 times and record the average accuracy with standard deviation. And we sample different label sets each time to mitigate the randomness. As can be seen, our model obtains the overall best results on all transfer tasks. Specially, TFGDA greatly surpasses the SOTA method SGDA [2] by $+7.3\%$ and $+10.0\%$ on "Micro-F1" and "Macro-F1" respectively for the **C→A** task, implying the superiority in extracting domain-invariant features. Notably, TFGDA enhances performance substantially on two hard transfer tasks, **C→A** and **D→A**,

Table 1: Transfer performance (%) on six transfer tasks with a source graph label rate of 5% for semi-supervised graph domain adaptation.

| Methods | A→C | | A→D | | C→A | | C→D | | D→A | | D→C | |
|---|---|---|---|---|---|---|---|---|---|---|---|---|
| | Micro-F1 | Macro-F1 | Micro-F1 | Macro-F1 | Micro-F1 | Macro-F1 | Micro-F1 | Macro-F1 | Micro-F1 | Macro-F1 | Micro-F1 | Macro-F1 |
| MLP [2] | $41.3_{\pm1.15}$ | $35.8_{\pm0.72}$ | $42.8_{\pm0.88}$ | $36.3_{\pm0.77}$ | $39.4_{\pm0.57}$ | $33.7_{\pm0.58}$ | $43.7_{\pm0.69}$ | $36.7_{\pm0.55}$ | $37.3_{\pm0.32}$ | $30.8_{\pm0.37}$ | $39.4_{\pm0.99}$ | $32.8_{\pm0.99}$ |
| GCN [30] | $54.4_{\pm1.52}$ | $52.0_{\pm1.62}$ | $56.9_{\pm2.33}$ | $53.4_{\pm2.81}$ | $54.1_{\pm1.40}$ | $52.3_{\pm1.98}$ | $58.9_{\pm0.99}$ | $54.5_{\pm1.55}$ | $50.1_{\pm2.14}$ | $48.0_{\pm3.28}$ | $56.0_{\pm1.24}$ | $51.9_{\pm1.49}$ |
| GSAGE [31] | $49.3_{\pm2.18}$ | $46.4_{\pm2.06}$ | $51.8_{\pm1.35}$ | $47.4_{\pm1.62}$ | $46.8_{\pm2.56}$ | $45.0_{\pm2.78}$ | $51.7_{\pm1.95}$ | $48.1_{\pm1.97}$ | $41.7_{\pm2.17}$ | $37.4_{\pm4.59}$ | $45.4_{\pm2.11}$ | $39.3_{\pm3.45}$ |
| GAT [32] | $55.1_{\pm3.22}$ | $50.8_{\pm1.45}$ | $55.3_{\pm2.52}$ | $51.8_{\pm2.60}$ | $50.0_{\pm1.20}$ | $45.6_{\pm2.36}$ | $55.4_{\pm2.73}$ | $49.2_{\pm2.59}$ | $44.8_{\pm2.74}$ | $38.3_{\pm4.84}$ | $50.4_{\pm3.35}$ | $42.0_{\pm4.46}$ |
| GIN [85] | $64.6_{\pm2.47}$ | $56.0_{\pm2.73}$ | $60.0_{\pm2.09}$ | $51.3_{\pm3.99}$ | $57.1_{\pm1.19}$ | $54.4_{\pm2.57}$ | $62.0_{\pm1.05}$ | $56.8_{\pm1.40}$ | $51.9_{\pm2.00}$ | $45.4_{\pm2.16}$ | $60.2_{\pm3.05}$ | $53.0_{\pm2.10}$ |
| DANN [86] | $44.3_{\pm2.03}$ | $39.3_{\pm1.86}$ | $44.0_{\pm1.42}$ | $38.7_{\pm1.47}$ | $41.8_{\pm1.95}$ | $37.6_{\pm1.24}$ | $45.5_{\pm0.71}$ | $39.6_{\pm1.55}$ | $37.8_{\pm3.66}$ | $33.2_{\pm2.23}$ | $41.7_{\pm2.32}$ | $35.6_{\pm2.55}$ |
| CDAN [12] | $44.6_{\pm1.30}$ | $38.6_{\pm1.07}$ | $45.5_{\pm0.85}$ | $38.0_{\pm0.86}$ | $42.4_{\pm0.64}$ | $36.2_{\pm1.17}$ | $46.7_{\pm1.17}$ | $39.2_{\pm0.96}$ | $39.0_{\pm1.08}$ | $32.3_{\pm1.09}$ | $41.7_{\pm1.55}$ | $34.8_{\pm1.56}$ |
| DANN$_{GCN}$ [2] | $63.0_{\pm6.75}$ | $59.6_{\pm6.02}$ | $62.2_{\pm1.90}$ | $57.7_{\pm3.16}$ | $56.7_{\pm0.38}$ | $55.2_{\pm1.03}$ | $65.3_{\pm2.04}$ | $59.0_{\pm2.39}$ | $52.3_{\pm2.59}$ | $48.6_{\pm4.52}$ | $58.1_{\pm2.78}$ | $52.4_{\pm3.81}$ |
| CDAN$_{GCN}$ [2] | $70.3_{\pm0.84}$ | $66.5_{\pm0.66}$ | $65.0_{\pm1.00}$ | $61.3_{\pm0.96}$ | $56.3_{\pm1.78}$ | $53.6_{\pm2.70}$ | $65.2_{\pm2.19}$ | $58.8_{\pm2.38}$ | $53.0_{\pm1.34}$ | $48.7_{\pm3.51}$ | $59.0_{\pm1.52}$ | $53.3_{\pm1.99}$ |
| UDA-GCN [75] | $72.4_{\pm2.75}$ | $65.2_{\pm6.51}$ | $68.0_{\pm6.38}$ | $64.3_{\pm7.12}$ | $62.9_{\pm0.33}$ | $62.2_{\pm1.44}$ | $71.4_{\pm2.56}$ | $67.5_{\pm2.25}$ | $55.8_{\pm3.50}$ | $52.4_{\pm2.68}$ | $65.2_{\pm4.41}$ | $60.7_{\pm6.84}$ |
| AdaGCN [1] | $70.8_{\pm0.95}$ | $68.5_{\pm0.73}$ | $68.2_{\pm3.84}$ | $64.2_{\pm3.91}$ | $61.5_{\pm2.20}$ | $60.4_{\pm3.15}$ | $69.1_{\pm1.96}$ | $65.8_{\pm2.87}$ | $56.1_{\pm1.75}$ | $53.8_{\pm2.95}$ | $64.1_{\pm0.91}$ | $62.8_{\pm1.56}$ |
| CoCo [27] | $72.7_{\pm1.36}$ | $66.8_{\pm1.15}$ | $68.3_{\pm2.31}$ | $64.1_{\pm2.68}$ | $62.7_{\pm0.95}$ | $61.5_{\pm1.18}$ | $71.6_{\pm1.76}$ | $67.3_{\pm1.93}$ | $56.7_{\pm1.47}$ | $54.1_{\pm1.29}$ | $66.0_{\pm0.88}$ | $64.4_{\pm1.13}$ |
| StruRW [22] | $72.9_{\pm1.21}$ | $67.1_{\pm1.07}$ | $68.5_{\pm0.94}$ | $64.4_{\pm1.03}$ | $63.6_{\pm1.05}$ | $61.9_{\pm1.19}$ | $71.8_{\pm2.06}$ | $67.6_{\pm2.45}$ | $57.0_{\pm1.72}$ | $54.2_{\pm1.38}$ | $65.7_{\pm0.96}$ | $63.1_{\pm1.25}$ |
| SGDA [2] | $75.6_{\pm0.57}$ | $71.4_{\pm0.82}$ | $69.2_{\pm0.73}$ | $64.7_{\pm2.36}$ | $66.3_{\pm0.68}$ | $62.3_{\pm0.96}$ | $72.9_{\pm1.26}$ | $68.9_{\pm1.83}$ | $60.6_{\pm0.86}$ | $56.0_{\pm0.90}$ | $73.2_{\pm0.59}$ | $69.3_{\pm1.01}$ |
| **TFGDA-S** | $55.8_{\pm1.76}$ | $53.6_{\pm1.84}$ | $54.2_{\pm2.11}$ | $44.9_{\pm2.04}$ | $58.2_{\pm1.52}$ | $48.9_{\pm1.94}$ | $57.0_{\pm1.03}$ | $46.3_{\pm1.60}$ | $49.8_{\pm2.33}$ | $41.0_{\pm3.46}$ | $55.9_{\pm1.41}$ | $45.2_{\pm1.72}$ |
| **TFGDA-T** | $72.6_{\pm0.55}$ | $66.3_{\pm0.83}$ | $65.9_{\pm0.85}$ | $62.4_{\pm1.39}$ | $64.3_{\pm0.88}$ | $61.8_{\pm0.76}$ | $65.6_{\pm0.97}$ | $54.7_{\pm1.24}$ | $56.2_{\pm0.78}$ | $53.4_{\pm0.87}$ | $68.5_{\pm0.44}$ | $67.4_{\pm0.92}$ |
| **TFGDA-D** | $75.8_{\pm0.38}$ | $70.7_{\pm0.69}$ | $71.2_{\pm0.61}$ | $67.5_{\pm1.15}$ | $68.5_{\pm0.57}$ | $63.7_{\pm1.02}$ | $72.2_{\pm0.88}$ | $68.1_{\pm1.13}$ | $63.1_{\pm0.74}$ | $58.5_{\pm0.79}$ | $73.4_{\pm0.51}$ | $71.1_{\pm0.88}$ |
| **TFGDA-R** | $74.4_{\pm0.46}$ | $70.1_{\pm0.77}$ | $68.8_{\pm0.54}$ | $64.7_{\pm0.98}$ | $65.6_{\pm0.49}$ | $62.4_{\pm0.85}$ | $69.7_{\pm0.94}$ | $63.3_{\pm1.22}$ | $62.7_{\pm0.60}$ | $56.8_{\pm0.83}$ | $72.1_{\pm0.43}$ | $69.6_{\pm0.85}$ |
| **TFGDA-TD** | $78.9_{\pm0.59}$ | $76.9_{\pm0.62}$ | $72.9_{\pm0.86}$ | $70.8_{\pm1.34}$ | $70.1_{\pm0.65}$ | $68.9_{\pm0.89}$ | $73.7_{\pm1.14}$ | $71.1_{\pm1.39}$ | $64.8_{\pm0.68}$ | $62.6_{\pm0.76}$ | $75.2_{\pm0.62}$ | $72.4_{\pm0.93}$ |
| **TFGDA-TR** | $78.4_{\pm0.64}$ | $75.8_{\pm0.48}$ | $72.3_{\pm0.60}$ | $68.3_{\pm1.13}$ | $70.5_{\pm0.58}$ | $69.6_{\pm0.95}$ | $73.4_{\pm0.93}$ | $70.8_{\pm1.52}$ | $64.2_{\pm0.71}$ | $61.8_{\pm0.81}$ | $76.3_{\pm0.54}$ | $72.6_{\pm0.83}$ |
| **TFGDA-DR** | $79.2_{\pm0.41}$ | $77.4_{\pm0.50}$ | $73.2_{\pm0.49}$ | $71.6_{\pm0.94}$ | $72.0_{\pm0.53}$ | $71.4_{\pm0.92}$ | $74.5_{\pm1.10}$ | $71.5_{\pm1.47}$ | $65.3_{\pm0.63}$ | $63.0_{\pm0.70}$ | $77.1_{\pm0.49}$ | $72.9_{\pm0.87}$ |
| **TFGDA** | $\mathbf{81.0}_{\pm0.34}$ | $\mathbf{78.9}_{\pm0.46}$ | $\mathbf{75.3}_{\pm0.51}$ | $\mathbf{73.2}_{\pm0.89}$ | $\mathbf{73.6}_{\pm0.61}$ | $\mathbf{72.3}_{\pm0.94}$ | $\mathbf{76.0}_{\pm1.02}$ | $\mathbf{72.6}_{\pm1.35}$ | $\mathbf{66.9}_{\pm0.59}$ | $\mathbf{64.3}_{\pm0.72}$ | $\mathbf{78.9}_{\pm0.47}$ | $\mathbf{74.4}_{\pm0.91}$ |

and achieves outstanding results in some easy transfer scenarios, such as **A→C** and **D→C**, implying that TFGDA can successfully minimize domain discrepancy. More importantly, the results with a smaller fluctuation range indicate not only the stability of our framework, but also its robustness in the face of different scenarios. Furthermore, we find that most competitors, such as CDAN$_{GCN}$, UDA-GCN, and AdaGCN, exhibit poor performance due to their limited ability to align domains and ineffective utilization of unlabeled nodes. In contrast, TFGDA effectively solves these challenges.

## 5.3 Analysis and Ablation Study

Due to the limitation of page size, more experiments and analysis are placed on the **Appendix**.

**1) Ablation Study:** To investigate the contribution of each component in TFGDA, we compare TFGDA and its 7 variants on various tasks. The variants of TFGDA are shown in Table 2, and the detailed ablation study results are gathered in Table 1.

**Contribution of Each Component:** The results in Table 1 reflect the following observations: **(1)** TFGDA-S (baseline) performs poorly on all tasks because it encounters overfitting problem. **(2)** Compared to TFGDA-S, variants TFGDA-T, TFGDA-D and TFGDA-R achieve significant performance gains, indicating that preserving topological structure information, reducing domain discrepancy on the sphere, and guiding discriminative clustering of unlabeled nodes all facilitate the learning of domain-invariant node features.

Table 2: Variants of TFGDA.

| Variant | $\mathcal{L}_{cls}$ | $\mathcal{L}_{stsa}$ | $\mathcal{L}_{sda}$ | $\mathcal{L}_{rnc}$ |
|---|---|---|---|---|
| TFGDA-S | ✓ | | | |
| TFGDA-T | ✓ | ✓ | | |
| TFGDA-D | ✓ | | ✓ | |
| TFGDA-R | ✓ | | | ✓ |
| TFGDA-TD | ✓ | ✓ | ✓ | |
| TFGDA-TR | ✓ | ✓ | | ✓ |
| TFGDA-DR | ✓ | | ✓ | ✓ |
| **TFGDA** | ✓ | ✓ | ✓ | ✓ |

**Correlation of Our Strategies:** As shown in Table 1, the combination of different strategies improves the model's transfer performance, implying a clear complementary relationship among the STSA, SDA, and RNC strategies.

**2) Visualization of Node Features:** To show the superior transfer ability of our model, we employ t-SNE to visualize node features on task **A→C** under the same 5% label rate setting, as depicted in Figure 2. Although the SOTA method SGDA reduces domain discrepancy to a certain, there are some overlaps between different clusters, causing some hard-to-transfer nodes to be easily misclassfied. In comparison, TFGDA achieves exactly 5 clusters with clean decision boundaries, indicating that our model can capture more fine-grained transferable features as well as align more complex distributions.

**3) Effect of STSA:** To showcase the effectiveness of preserving topological structure information in assisting GTL, we conduct in-depth experiments from both quantitative and visual aspects: **(1)** As depicted in Table 1, TFGDA-TD and TFGDA-TR greatly outperform TFGDA-D and TFGDA-R respectively, indicating that aligning the topological structures of the input and latent spaces can effectively boost the model's generalization. **(2)** As shown in Figure 2, compared to TFGDA-R,

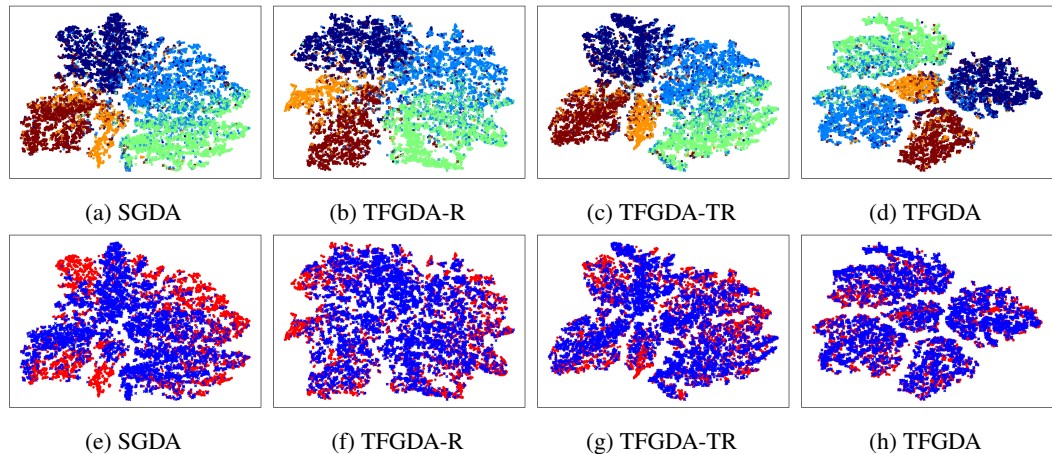

(a) SGDA        (b) TFGDA-R        (c) TFGDA-TR        (d) TFGDA

(e) SGDA        (f) TFGDA-R        (g) TFGDA-TR        (h) TFGDA

Figure 2: The t-SNE visualization of representations learned by SGDA, TFGDA and its two variants on **A→C** task with 5% label rate. In all subfigures, the marks ● and × represent the source domain and target domain, respectively. **Fig 2(a-d)** depict category alignment (Different colors denotes different classes). **Fig 2(e-h)** depict domain alignment (Red: Source domain; Blue: Target domain).

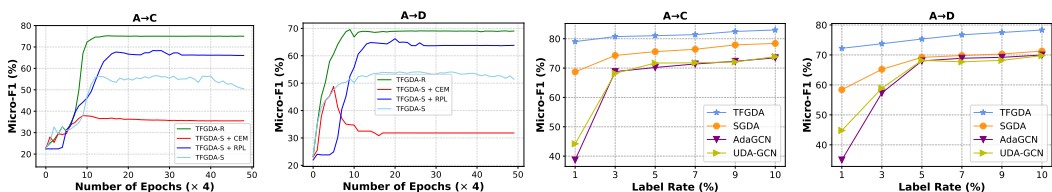

Figure 3: The trend of Micro-F1 during training.    Figure 4: Performance with different label rates.

variant TFGDA-TR exhibits better intra-class compactness and inter-class separability in feature space, implying that the incorporation of topological structure information helps guide the discriminative clustering of unlabeled nodes, thereby promoting the learning of domain-invariant nodes features.

**4) Effect of RNC:** To show the effectiveness of our RNC strategy, we compare it with existing node clustering strategies, including conditional entropy minimization strategy (**CEM**) [75] and the recently proposed re-weighted pseudo-labeling strategy (**RPL**) [2]. We investigate the trend of Micro-F1 score during training on tasks **A→C** and **A→D**. The curves in Figure 3 reflect the following observations: **(1)** TFGDA-R converges more smoothly and quickly, achieving higher transfer performance, which suggests that RNC strategy can effectively accelerate the learning of domain-invariant features. **(2)** Compared to TFGDA-S (baseline), TFGDA-S + CEM suffers from severe performance degradation, as CEM may enforce over-confident probability on some misclassified unlabeled nodes. **(3)** RPL strategy fails to achieve satisfactory performance as it's sensitive to pseudo-label noise.

**5) Effect of Label Rate:** To verify the model's robustness under different label scarcity settings, we evaluate the performance of different methods on tasks **A→C** and **A→D**, using the following label rates for the source graph: 1%, 5%, 7%, 9%, and 10% respectively, as shown in Figure 4. It can be observed that our TFGDA significantly outperforms other competitors, even in the most challenging environment of 1% label rate, indicating the superiority of TFGDA in capturing transferable features.

## 6 Conclusion

In this paper, we develop a novel model named TFGDA for SGDA. Specially, we propose a STSA strategy to encode critical structure information into latent space, significantly improving model's transfer performance. Moreover, to stably reduce domain discrepancy, the SDA strategy is introduced to align features distributions on sphere. We also devise the RNC strategy to guide the clustering of unlabeled nodes to address the overfitting issue, greatly enhancing the model's robustness. Comprehensive experiments and analysis verify the superiority of our TFGDA.

# 7 Acknowledgements

This work was supported in part by the National Natural Science Foundation of China under Grants (No.62302098, No.62401355), and the Start-up Program for New Young Teacher of Shanghai Jiao Tong University (KJ3-0221-22-6349).

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

# A Appendix

## A.1 Datasets

We run experiments on three real-world graphs: *ACMv9* (**A**), *Citationv1* (**C**), and *DBLPv7* (**D**). In these graphs, every node refers to a paper, and the attribute of each paper is represented as a sparse bag-of-words vector derived from its title. The edges in these graphs depict citation relationships among the papers. Each node is assigned a five-class label, determined by its relevant research areas, including *Artificial Intelligence*, *Computer Vision*, *Database*, *Information Security*, and *Networking*.

Table 3 displays various statistical information of three graphs, including graph scale, attributes, average degree, and label proportion. We can observe substantial intrinsic discrepancy among these graphs. In this paper, we adopt an alternating approach where we select one of these graphs as the source domain, while considering the remaining two as the target domains.

Table 3: The statistics of three real-world graphs. Note that '#' means 'the number of'. 'Attr.' refer to 'Attributes'. 'Avg.' represents 'Average'.

| Graph | #Nodes | #Edges | #Attr. | Avg. Degree | Label Proportion (%) |
|---|---|---|---|---|---|
| *ACMv9* (**A**) | 9,360 | 15,602 | 5,571 | 1.667 | 20.5/29.6/22.5/8.6/18.8 |
| *Citationv1* (**C**) | 8,935 | 15,113 | 5,379 | 1.691 | 25.3/26.0/22.5/7.7/18.5 |
| *DBLPv7* (**D**) | 5,484 | 8,130 | 4,412 | 1.482 | 21.7/33.0/23.8/6.0/15.5 |

## A.2 Implementation Details

Our experiments are implemented using Pytorch library. Following previous work [2], we choose a two-layer GCN as the feature extractor $\mathcal{F}$ of TFGDA model. We randomly initialize the shift parameter $\xi^s$ and $\xi^t$ using uniform distributions $\mathcal{U}_s(-\epsilon, \epsilon)$ and $\mathcal{U}_t(-\epsilon, \epsilon)$, respectively. In all experiments, we set the value of $\epsilon$ to 0.5. For all transfer tasks, we perform each random experiment 5 times and record the average accuracy with standard deviation. And we sample different label sets for each experiment to mitigate the randomness.

To optimize the network, we employ Adam optimizer with a weight decay of 0.001 for better convergence. The base learning rate is set to 0.002 for all tasks. In terms of the balanced coefficients $\varepsilon$ and $\eta$, we choose $\varepsilon = 0.3$ and $\eta = 1$ for all transfer tasks. Furthermore, instead of fixing the trade-off parameter $\lambda$ of RNC strategy, we adopt a progressive schedule as [2] to dynamically adjust $\lambda$ from 0 to 1 by multiplying by $\frac{\theta}{\Theta}$ to more stably guide the discriminative clustering of unlabeled nodes, where $\theta$ is the current epoch and $\Theta$ is the maximum epoch. Parameter $\tau$ in Eq.(8) is set to 0.999, and the spherical radius $r$ is set to the lower bound.

In our STSA strategy, we choose to retain 0-dimensional topological information in PDs. This is because some preliminary experiments have shown that using higher-dimension topological information does not lead to clear accuracy improvements but noticeably increases the model's training time. For the subgraph sampling operation, we set the number of subgraphs to 10 and the number of nodes in each subgraph to 800 to strike a balance between model performance and training efficiency. During the inference process, we disable the shift parameters branch in our RNC strategy. Notably, for all methods (including the compared methods), the dimension of node features is set to 512.

## A.3 More Experiments and Analysis

### A.3.1 More Ablation Study

**6) Effectiveness of SDA:** To demonstrate the effectiveness of SDA strategy, we compare it with existing domain alignment strategies, including adversarial training alignment strategy (**AT**) [1], sliced Wasserstein distance-based alignment strategy (**SWD**)[87], class-conditional MMD strategy (**CMMD**) [3], and the recently proposed shifting-guided adversarial training alignment strategy (**SAT**) [2]. We employ variant TFGDA-S as the baseline and evaluate the performance gains brought by these strategies on two typical transfer tasks: **A**→**C** and **A**→**D**.

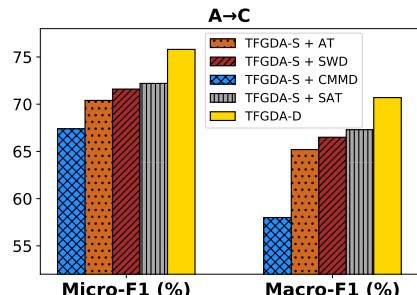
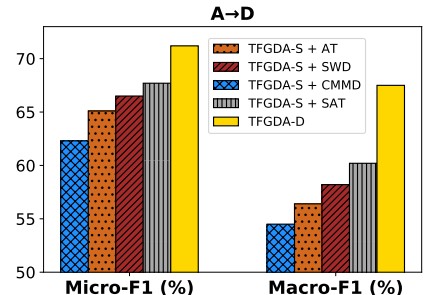

Figure 5: Transfer performance with different domain alignment strategies **A→C** and **A→D**.

Table 4: Notations.

| Notation | Description |
|---|---|
| $a$ | number of sampled subgraphs $\left\{\hat{\mathcal{G}}_1, \hat{\mathcal{G}}_2, \cdots, \hat{\mathcal{G}}_a\right\}$ |
| $b$ | number of great circles projections |
| $\varpi$ | a coefficient for controling the scale of feature perturbation |
| $\eta, \varepsilon, \lambda$ | balance parameters of loss terms $\mathcal{L}_{sda}, \mathcal{L}_{stsa}$ and $\mathcal{L}_{rnc}$ |
| $|\hat{\mathcal{V}}_i|$ | number of nodes in each subgraph |

The results shown in Figure 5 reflect the following observations: **(1)** Compared to adversarial training-based strategies such as AT and SAT, our SDA strategy can bring greater performance improvements to the model, especially in the hard transfer task **A→D**. This is because SDA implements a more stable distributions alignment process, which promotes the model to capture more transferable features. **(2)** The proposed SDA strategy works better than the SWD strategy, as SDA takes into account the manifold structure of the data in the spherical space, while SWD primarily focuses on the distribution discrepancy in the Euclidean space. **(3)** TFGDA-D significantly outperforms all compared strategies, which indicates the superiority of our SDA strategy in mitigating domain discrepancy and extracting domain-invariant features.

### A.3.2  Parameter Sensitivity

The key parameters and notations in our TFGDA model are summarized in Table 4.

**7) Effect of Subgraphs:** As depicted in Figure 6, we investigate the effect of different numbers of sampled subgraphs $a$ and different subgraph sizes $|\hat{\mathcal{V}}_i|$ on the model's transfer performance, respectively.

We can obtain the following observations: **(1)** As the number of sampled subgraphs $a$ increases, the model's transfer performance gradually improves until it converges. However, a large number of subgraphs can lead to slow model training. Therefore, in our model, we set the number of subgraphs to 10 to balance model's accuracy and training efficiency; **(2)** Similarly, as the size of the subgraphs $|\hat{\mathcal{V}}_i|$ increases, the model's accuracy gradually improves until it converges. However, excessively large subgraphs also significantly increase training time. Hence, to strike a balance between performance and training efficiency, we set the number of nodes in each subgraph to 800. **(3)** When the number $a$ or size $|\hat{\mathcal{V}}_i|$ of the subgraphs is set too small, the model's performance becomes less robust, exhibiting a higher standard deviation. This is because, in such cases, the subgraphs struggle to capture the structure information present in the original graph sufficiently.

**8) Effect of Feature Perturbation Scale:** The constraint coefficient $\varpi$ of shift parameters $\xi_s$ and $\xi_t$ is responsible for controlling the scale of node feature perturbation. To investigate the effect of coefficient $\varpi$ in our framework, we train our TFGDA models with different $\varpi$ and evaluate their performance on two tasks: **A→C** and **A→D**.

The results are illustrated in Figure 7 As $\varpi$ increase, the model's accuracy first rises and then falls, which implies that properly perturbing the node feature can effectively enhance the model's robustness.

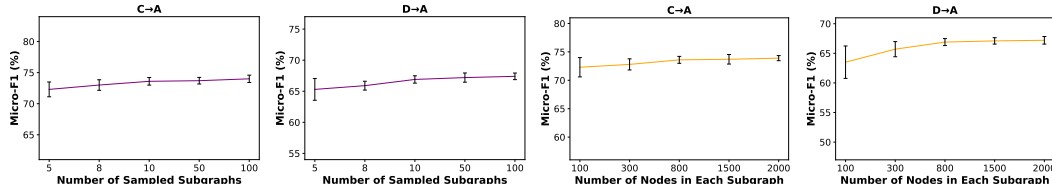

Figure 6: Transfer performance with subgraphs at different scales on **C**→**A** and **D**→**A** tasks.

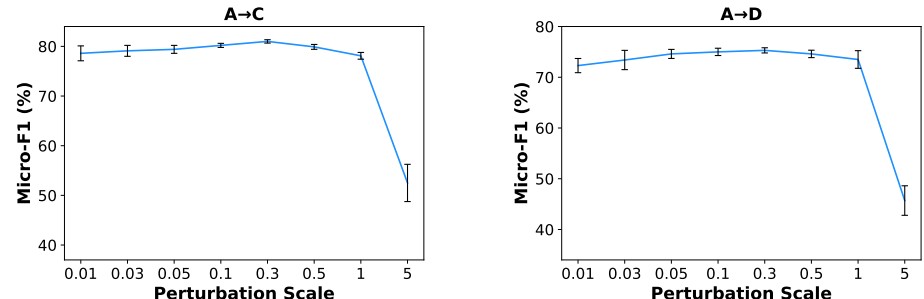

Figure 7: Transfer performance with different feature perturbation scale $\varpi$ on **A**→**C** and **A**→**D** tasks.

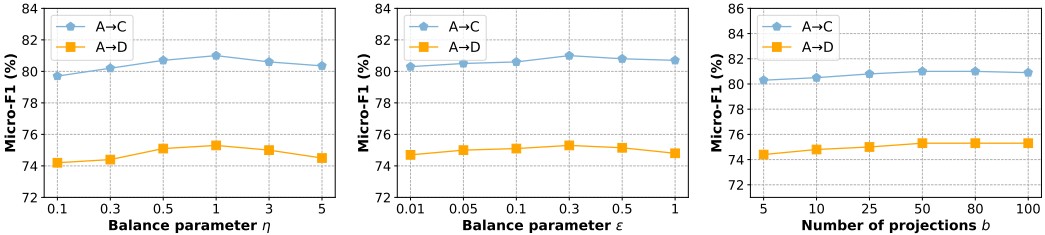

Figure 8: Parameter sensitivity analyses of parameters $\eta$, $\varepsilon$, and $b$ on **A**→**C** and **A**→**D** tasks.

Specially, when $\varpi \in [0.05, 0.5]$, our model can achieve stable transfer performance with a very small standard deviation. However, when the coefficient $\varpi$ is set to a large value, unlabeled nodes struggle to cluster in the correct direction, leading to a significant degradation in model performance.

**9) Effect of Hyper-parameter:** Figure 8 depicts our evaluation of the sensitivity of several hyper-parameters on transfer tasks **A**→**C** and **A**→**D**. The evaluated hyper-parameters include balance parameters $\eta$ and $\varepsilon$, and the number of projections $b$. For the parameter $\eta$, we find that selecting an appropriate value to adjust the SDA loss $\mathcal{L}_{sda}$ can effectively decrease domain discrepancy. Moreover, we can observe that our model is robust to changes in parameter $\varepsilon$, indicating the stability of our STSA strategy. The results also implies that our SDA strategy maintains stability when aligning feature distributions across domains, regardless of variations in the specific value of the parameter $b$. When $b$ is extremely small, accurately approximating the SSW distance (*i.e.*, domain discrepancy Eq.(12)) in the SDA strategy becomes difficult, leading to a slight decrease in model's transfer performance.

### A.3.3 More Analysis

**10) Why our RNC strategy can avoid degenerate clustering solutions ?:** The mutual information (MI) objective function $\mathcal{L}_{mi}(\mathbb{Q})$ in Eq. 15 can be expanded to: $\mathcal{L}_{mi}(\mathbb{Q}) = \mathcal{L}_{mi}(\phi, \phi^{\xi}) = E(\phi) - E(\phi|\phi^{\xi})$ [73, 74]. Therefore, maximizing this objective $\mathcal{L}_{mi}$ involves a trade-off between minimizing the conditional cluster assignment entropy $E(\phi|\phi^{\xi})$ and maximizing the entropy of individual cluster assignments $E(\phi)$. Specially, the minimum value of $E(\phi|\phi^{\xi})$ is 0, which is achieved when the cluster assignments can be precisely predicted from each other. The maximum value of $E(\phi)$ is $\ln K$, achieved when all clusters have an equal probability of being selected, where $K$ is the number of classes. This situation arises when the data is evenly distributed among the clusters, resulting in an equal distribution of their masses. Hence, the loss function cannot minimized when all

samples are assigned to the same cluster. In such case, maximizing MI naturally achieves a balance between reinforcing predictions and equalizing the cluster masses, thereby avoiding the occurrence of degenerate clustering solutions. The t-SNE node features visualization results in the main text also validate this viewpoint.

### A.3.4 Limitation

Our model might has the following two limitations:

**(i)** As depicted in Figure 6 of the Appendix, the effectiveness of our STSA strategy depends on the quality of the subgraphs (*i.e.*, subgraph size $|\hat{\mathcal{V}}_i|$ and the number of subgraphs $a$). As a result, the model's transfer performance may decrease if the subgraph size is small or the number of subgraphs is small, as these local subgraphs are difficult to sufficiently capture the properties of the original graph.

**(ii)** As depicted in Figure 7 of the Appendix, the transfer ability of the model is affected by the feature perturbation scale $\varpi$ in the RNC strategy. Specially, when the feature perturbation scale $\varpi$ is set to a large value, it is difficult for the RNC strategy to accurately capture the intrinsic invariant features of nodes, which affects the discriminative clustering of unlabeled nodes and leads to a decline in model's generalization performance.

