# OpenReview forum: "TFGDA: Exploring Topology and Feature Alignment in Semi-supervised Graph Domain Adaptation through Robust Clustering"
_NeurIPS.cc/2024/Conference — NeurIPS 2024 poster_

### Official Review · Reviewer_R5oT · 2024-06-25

**Soundness:** 3
**Presentation:** 3
**Contribution:** 3
**Rating:** 7
**Confidence:** 5

**Summary:**

This paper proposes a novel framework named TFGDA for the semi-supervised graph domain adaptation. Considering that existing graph domain adaptation works overlook the utilization of graph structure information, this paper innovatively proposes a STSA strategy that fully leverages graph structure information to promote the learning of domain-invariant node features. Moreover, TFGDA also devises a SDA strategy to achieve a fine-grained feature distributions alignment, which maps node features onto multiple great circles and utilizes the spherical sliced-Wasserstein distance to quantify the domain discrepancy. To remedy the overfitting phenomenon caused by limited source labeled nodes, a novel RNC strategy is proposed to guide the discriminative clustering of unlabeled nodes. Comprehensive experiments on various benchmarks demonstrate that the proposed TFGDA outperforms state-of-the-art methods in semi-supervised graph domain adaptation tasks.

**Strengths:**

1) This paper is well-motivated and easy-to-follow. The overall structure is clear. Source graph label scarcity is indeed a common challenge encountered in real-world applications.

2) Preserving the graph structure from a topological perspective to promote the extraction of transferable node feature is a novel idea. It effectively utilizes the inherent properties of the graph to enhance model’s transfer performance, which can shed some light on future research on graph transfer learning.

3) It is a novel attempt to utilize the SDA strategy to achieve fine-grained feature distribution alignment in spherical space.

4) The motivation behind the RNC strategy is clear. The experimental results also validate that the RNC strategy effectively mitigates overfitting issues and significantly  outperforms other strategies.

5) The effectiveness and SOTA transfer performance of TFGDA has been confirmed through extensive ablation studies, visualization results, and comparative experiments.

**Weaknesses:**

I have several concerns and questions as below:

[About writing]
1) It is suggested to include some discussions on practical applications of graph transfer learning in the Sec 1, which may better emphasize the significance of the research.

[About the proposed method]
2) The proposed STSA strategy employ persistent homology to extract the topological structures of both graphs. In topological theory, topological features of different dimensions capture the structure of the underling space from different perspectives. What dimensions of topological features did the authors use in STSA strategy to align the topological structures of the input space and latent space? How was this determined?

3) What distance metric was used when constructing the Vietoris-Rips simplicial complex in the STSA strategy?

4) Previous domain adaptation works [1] have used the K-means algorithm to guide unlabeled nodes clustering. What advantages does the RNC strategy have compared to K-means?

[About the experiments]
5) Although the authors conducted detailed experiments on mainstream graph domain adaptation datasets (i.e., ACMv9, Citationv1 and DBLPv7), I hope they discuss the scalability of TFGDA on larger graph datasets, such as the recommender systems dataset AliCVR [2].

6) It is suggested to compare and discuss the proposed TFGDA with the recently introduced GIFI method [3].

Ref [1] Deng W, Liao Q, Zhao L, et al. Joint clustering and discriminative feature alignment for unsupervised domain adaptation[J]. IEEE Transactions on Image Processing, 2021, 30: 7842-7855.
Ref [2] Guo G, Wang C, Yan B, et al. Learning adaptive node embeddings across graphs[J]. IEEE Transactions on Knowledge and Data Engineering, 2022, 35(6): 6028-6042.
Ref [3] Qiao Z, Xiao M, Guo W, et al. Information filtering and interpolating for semi-supervised graph domain adaptation[J]. Pattern Recognition, 2024, 153: 110498.

**Questions:**

See Weakness.

**Limitations:**

See Weakness. If the authors address all my concerns, I am pleased to improve the final score.

---

> ### Author Rebuttal · Authors · 2024-08-06
>
> We appreciate Reviewer R5oT for the thorough review of the manuscript and the valuable comments that will aid in enhancing our paper.
>
> **Q1: Include some discussions on practical applications of graph transfer learning in the Sec.1.**
> **A1:** Graph Transfer Learning can be applied in various practical scenarios, such as recommender systems [K1], molecular property prediction [K2], social and academic networks analysis [K3], cross-modal retrieval [K4], and human parsing [K5].
>
> Ref.[K1] Learning adaptive node embeddings across graphs. TKDE 2022.\
> Ref.[K2] Transfer learning with graph neural networks for improved molecular property prediction. NC, 2024.\
> Ref.[K3] A comprehensive survey on graph neural networks. TNNLS, 2020.\
> Ref.[K4] Cross-Domain Transfer Hashing for Efficient Cross-modal Retrieval. TCSVT, 2024.\
> Ref.[K5] Graphonomy: Universal Human Parsing via Graph Transfer Learning. CVPR 2019.
>
> **Q2: What dimensions of topological features did the authors use in STSA strategy? How was this determined?**
> **A2:** As described in Section A.2 of the Appendix (lines 508-511), we use 0-dimensional topological features in our STSA strategy to align the topological structures of the input and latent spaces. Because some preliminary experiments have shown that using higher-dimension topological information does not lead to clear accuracy improvements but noticeably increases the model’s training time.\
> Additionally, the topological theory [K6-K7] has demonstrated that low-dimensional topological features (e.g., 0-dimensional) can roughly reflect the topological structure of the data, while high-dimensional topological features (e.g., 3-dimensional and 4-dimensional) are responsible for capturing some complex details of topological structure.
> Thus, to balance computational efficiency and model performance, we choose to retain 0-dimensional topological features in our STSA strategy.
>
> Ref.[K6] Computing persistent homology. 2004.\
> Ref.[K7] Persistent homology-a survey. 2008.
>
> **Q3: What distance metric was used when constructing the Vietoris-Rips simplicial complex in the STSA strategy?**
> **A3:** We use Euclidean distance as the distance metric to construct the Vietoris-Rips simplicial complex in our STSA strategy.
>
> **Q4: What advantages does the RNC strategy have compared to the K-means algorithm?**
> **A4:** Compared to the K-means clustering algorithm, our RNC strategy has two main advantages:
>
> **1)** Using the K-means algorithm in neural networks results in a two-stage clustering process [K8]. Specifically, the deep features extracted by the network are first clustered using the K-means algorithm to generate pseudo-labels. These pseudo-labels are then used to guide the model training. This clustering process inevitably introduces pseudo-labels noise into the model during training, limiting the model's generalization ability. On the contrary, our proposed RNC strategy is an end-to-end clustering method that does not involve any pseudo-labels, thereby enhancing the model's robustness in face of different transfer scenarios.
>
> **2)** When combined with representation learning, the K-means algorithm is prone to degenerate clustering solutions [K8-K10], where all unlabeled nodes are assigned to the same cluster, severely affecting the construction of decision boundaries.
> The proposed RNC strategy is designed based on the mutual information theory, which effectively guides unlabeled nodes toward robust clustering and naturally avoids degenerate clustering solutions. As shown in Section A.3.3 of the Appendix (lines 566-579), we have provided a detailed theoretical analysis that explains how our RNC strategy can avoid degenerate clustering solutions. Furthermore, as mentioned in Section 4.5 (lines 234-236), the experimental results in Figure 2 and Figure 3 have also validated this point.
>
> Ref. [K8] Deep Clustering for Unsupervised Learning of Visual Features. ECCV 2018.\
> Ref. [K9] On strategies to fix degenerate k-means solutions. Journal of Classification 2017.\
> Ref. [K10] How to Use K-means for Big Data Clustering. PR 2023.
>
> **Q5: Discuss the scalability of TFGDA on larger graph datasets, such as the AliCVR dataset.**
> **A5:** Based on your valuable suggestion, we follow Ref.[K1] to utilize a real-world large-scale recommender systems dataset Ali-CVR to conduct additional experiments. Specially, Ali-CVR can be divided into 3 large-scale graphs: **AN**, **A11** and **AB11**.
> We report the node classification accuracy in the target graph, and the results in Table R1 demonstrate the scalability and stability of our method.
>
> **Table R1: Node classification accuracy (%) in the target graph. Source graph label rate: 10\%. [Source graph $\rightarrow$ Target graph]**
> | Method | **AN**$\rightarrow $**A11** | **AN**$\rightarrow $**AB11** | **A11**$\rightarrow $**AN** |
> | ------ | ------ | ------ | ------ |
> | UDA-GCN | 20.66$\_\{\pm2.41}$ | 23.47$\_\{\pm3.07}$ | 21.29$\_\{\pm1.75}$ |
> | GraphAE | 23.80$\_\{\pm2.06}$ | 27.19$\_\{\pm2.53}$ | 25.04$\_\{\pm1.58}$ |
> | SGDA | 27.42$\_\{\pm1.83}$ | 31.25$\_\{\pm2.20}$ | 28.36$\_\{\pm1.34}$ |
> | **TFGDA** | **33.67**$\_\{\pm1.42}$ | **35.82**$\_\{\pm1.73}$ | **34.15**$\_\{\pm1.09}$ |
>
> **Q6: It is suggested to compare and discuss the TFGDA with the GIFI method.**
> **A6:** Following your suggestion, we compare the transfer performance of our method with the GIFI on the challenging transfer scenario, where only 5% of the source graph nodes are labeled.  The results in the Table R2 demonstrate the effectivenss of our method.
>
> **Table R2: Transfer performance (%) measured by Macro-F1 on some transfer tasks. Source graph label rate: 5%.**
> | Method | **A**$\rightarrow $**C** | **C**$\rightarrow $**D**  | **D**$\rightarrow $**A** |
> | ------ | ------ | ------ | ------ |
> | GIFI | 73.7$\_\{\pm0.69}$ | 69.0$\_\{\pm2.05}$ | 64.1$\_\{\pm1.08}$ |
> | **TFGDA** | **78.9**$\_\{\pm0.46}$ | **72.6**$\_\{\pm1.35}$ | **64.3**$\_\{\pm0.72}$ |

---

> ### Author Response · Authors · 2024-08-12
> **Response to Reviewer R5oT**
>
> Dear Reviewer R5oT:
>
> We thank your response and appreciation of our work and rebuttal. We will make sure to incorporate the new results and discussions into our revision to enable it to be a high-quality paper.
>
> Best Regards, Authors of 504.

---

### Official Review · Reviewer_byDA · 2024-06-29

**Soundness:** 3
**Presentation:** 3
**Contribution:** 3
**Rating:** 7
**Confidence:** 5

**Summary:**

This paper focus on the semi-supervised graph domain adaptation, and introduces a new framework called TFGDA. Graph usually contains complex structure information, while existing GTL studies often overlooks the importance of structure information when extracting transferable node features. TFGDA thus proposes a novel STSA strategy to utilize the topological structures information between input and latent spaces to assist GTL. To solve the instability caused by adversarial training-based domain adaptation methods, this paper also presents an SDA strategy to reduce cross-domain node feature distributions discrepancy in the spherical space. Furthermore, an innovative mutual information-based RNC strategy is proposed to address the overfitting issue by guiding the robust clustering of unlabeled target graph nodes. Extensive experimental results show that TFGDA outperforms existing state-of-the-art methods across various transfer learning tasks, indicating its superiority and stability.

**Strengths:**

(1) The paper is well-written and has a clear structure. Compared to the widely studied unsupervised domain adaptation in GTL, the semi-supervised domain adaptation is more relevant to real-world application scenarios. Therefore, it is meaningful to explore effective solutions to address the challenges faced by semi-supervised domain adaptation. In general, the paper is quite novel and worth reading.

(2) The introduction of the STSA strategy is well-motivated. Leveraging the graph structure information to facilitate graph transfer learning is indeed an innovative attempt and shows significant transfer performance gains on multiple tasks.

(3) The SDA strategy exhibits significant superiority over existing methods in reducing node feature distributions difference. Additionally, it is an interesting idea to devise a node clustering strategy RNC from the view of mutual information. Detailed experimental results demonstrate the effectiveness of RNC in addressing the overfitting problem and enhancing model robustness.

(4) The paper provides comprehensive experiment on multiple benchmark datasets to validate the superior transfer performance over existing state-of-the-art methods.

**Weaknesses:**

To make this paper more comprehensive, there are some concerns that I would like the authors to address.
(1)	This paper mainly utilizes multiple real-world academic graphs as datasets. Further exploration can be conducted on other types of graph datasets (such as real-world graph datasets), which can more effectively validate the generalizability of the proposed method.
(2)	Furthermore, I recommend the authors to include a discussion and comparison of the model’s inference efficiency in the paper.
(3)	[Minor comment:] While the t-SNE visualization results in Figure 2 clearly show the advantage of TFGDA in reducing feature distribution discrepancy, I recommend the authors to include additional quantitative metrics to better demonstrate the transfer ability of the method, such as the {A}-distance [1].
(4)	[Minor comment:] Since the SDA strategy contains some complex mathematical details, it is recommended to add a high-level algorithm table to summarize it.

Reference:
[1] Analysis of representations for domain adaptation. (Neurips 2006).

**Questions:**

I have some doubts and hope the authors to clarify them.
(1)	When the model performs inference on the target domain graph, is the shift parameter perturbation branch $\xi$ of RNC activated? Does the graph data flow through the regular branch or the perturbation branch before being fed into the classifier \mathcal{C} for inference?

(2)	What is the intrinsic reason for the need to introduce structure information in graph transfer learning frameworks? Is it because the structure information of the data tends to be lost as the network layers go deeper?

(3)	Some previous images based-transfer learning works [2-4] have utilized the wasserstein or the sliced wasserstein as distance metrics to minimize domain discrepancy. In the SDA strategy, why is the SSW distance better than these two distance in measuring domain differences?

Reference:
[2] Sliced wasserstein discrepancy for unsupervised domain adaptation. (CVPR 2019).
[3] Reliable weighted optimal transport for unsupervised domain adaptation. (CVPR 2020).
[4] Deepjdot: Deep joint distribution optimal transport for unsupervised domain adaptation. (ECCV 2018).

**Limitations:**

The authors have provided clear explanations of the limitations that the proposed method may encounter in the supplementary materials. This article does not have any potential negative societal impact.

---

> ### Author Rebuttal · Authors · 2024-08-06
>
> We thank the Reviewer byDA for the careful reading of the manuscript and the related comments, which are helpful to improve our paper.
>
> **W1: Further exploration can be conducted on other types of graph datasets.**
> **A1:** Based on your suggestion, we conduct additional experiments on a real-world recommender systems graph dataset called AliCVR to validate the generalizability of our method. Please refer to the answer **A5** to Reviewer **R5oT**.
>
> **W2: Include a discussion and comparison of the model’s inference efficiency.**
> **A2:** We compare model's inference time on the transfer task **A**$\rightarrow$**C** in the following Table R3. Due to the addition of the spherical space $\mathbb{S}\_\{r}^{d-1}$, our TFGDA exhibits a slightly longer inference time (1.02x) than the SOTA competitor SGDA [S1], which does not significantly increase the inference time but brings a clear performance gain.
> Additionally, it is worth mentioning that if the spherical space $\mathbb{S}\_\{r}^{d-1}$ is removed, during inference the network architecture of our TFGDA is almost identical to that of SGDA, as we will disable the shift parameters perturbation branch $\xi\_\{s}$ and $\xi\_\{t}$.
>
> **Table R3: Model's Inference time on Citationv1 (C).**
> | Transfer Task | Method | Inference Time on **C** | Micro-F1 |
> | ------ | ------ | ------ | ------ |
> | **A**$\rightarrow$**C** | SGDA | 5.46s | 75.6$\_\{\pm0.57}$ |
> |  | **TFGDA** | 5.58s | 81.0$\_\{\pm0.34}$ |
>
> Ref. [S1] Semi-supervised Domain Adaptation in Graph Transfer Learning. IJCAI 2023.
>
> **W3: Include additional quantitative metrics to better demonstrate the transfer ability of the method, such as the  $\mathcal{A}$-distance.**
> **A3:** Following your advice, we utilize the $\mathcal{A}$-distance as a metric to evaluate the transfer ability of different methods in Figure 2. Table R4 shows the results of the $\mathcal{A}$-distance on **A**$\rightarrow$**C** task. Since our method can accurately align feature distributions of two domains at the class-level, it achieves the smallest $\mathcal{A}$-distance.
>
> **Table R4: $\mathcal{A}$-distance on A$\rightarrow$C task.**
> | Method | $\mathcal{A}$-distance |
> | ------ | ------ |
> | TFGDA-R | 1.67 |
> | SGDA | 1.54 |
> | TFGDA-TR | 1.48 |
> | **TFGDA** | 1.36 |
>
> **W4: Add a high-level algorithm table to summarize the SDA strategy.**
> **A4:** Thanks for the valuable advice. We will include a a high-level algorithm table for the SDA strategy in the revised version to make the paper more comprehensive.
>
> **Q1: During inference, is the shift parameter perturbation branch $\xi$ of RNC activated? Does the graph data flow through the regular branch or the perturbation branch before being fed into the classifier $\mathcal{C}$ for inference?**
> **A5:** As mentioned in Section A.2 of the Appendix (line 513), during the inference process, we will disable the shift parameters perturbation branches $\xi_{s}$ and $\xi_{t}$.
> Thus, during the inference process, the graph data will be fed through the regular branch (i.e., without any feature perturbation) into the classifier $\mathcal{C}$ for the final prediction.
>
> **Q2: What is the intrinsic reason for the need to introduce structure information in graph transfer learning frameworks? Is it because the structure information of the data tends to be lost as the network layers go deeper?**
> **A6:** Notably, unlike images and time series data, graph data usually contains rich structure information that encodes complex relationships among nodes and edges. Most existing graph transfer learning works  directly use graph convolutional networks (GCNs)-based feature extractors to learn transferable node features. However, recent studies [S2-S4] have indicated that GCNs are insufficient in capturing the sophisticated structure information in graph, which means that the graph structure information may be lost or destroyed after passing through the GCNs-based feature extractor. Therefore, we propose the STSA strategy to encode these intrinsic structure information in graph (i.e., from the input space) into the latent space by aligning the topological structures of the input space and the latent space, effectively improving the model's transfer performance.
>
> Ref. [S2] Coco: A coupled contrastive framework for unsupervised domain adaptive graph classification. ICML 2023.\
> Ref. [S3] Graph Kernel Neural Networks. TNNLS 2024.\
> Ref. [S4] Theoretically improving graph neural networks via anonymous walk graph kernels. WWW 2021.
>
> **Q3: In the SDA strategy, why is the SSW distance better than the wasserstein and the sliced wasserstein distance metrics in measuring domain differences?**
> **A3:** **1)** As described in lines 182-184, due to numerous nodes in the graph (e.g., ACMv9 dataset has over 9000 nodes), directly using the classical Wasserstein distance to compute feature distributions discrepancy is computationally expensive, rendering it impractical for graph transfer learning task.
>
> **2)** To eliminate the negative influence of feature norm discrepancy on the learning of transferable node features, our SDA strategy thus guides feature distributions alignment in spherical space.  As described in lines 528-530, the sliced Wasserstein distance focuses on calculating discrepancy between distributions in Euclidean space [S5], making it difficult to precisely measure feature distributions discrepancy in spherical space. While our SSW distance fully considers the manifold structure of data in spherical space when assessing distribution discrepancy [S6], thereby facilitating a more precise quantification of difference in feature distributions. Furthermore, the detailed ablative experiments results in Figure 5 of the Appendix (lines 518-532) have already validated that our SSW distance is superior to the sliced Wasserstein distance in measuring domain discrepancy.
>
> Ref.[S5] Sliced Wasserstein Kernels for Probability Distributions. CVPR 2016.\
> Ref.[S6] Spherical Sliced-Wasserstein. ICLR 2023.

---

> ### Author Response · Authors · 2024-08-12
> **Looking forward to the reply**
>
> Dear Reviewer byDA:
>
> Thanks so much again for the time and effort in our work. According to the comments and concerns, we conduct the corresponding experiments and further discuss the related points. As the discussion period is nearing its end, please feel free to let us know if there are any other concerns. We would be happy to provide more details and clarifications.
>
> Best Regards, Authors of 504.

---

> > ### Comment · Reviewer_byDA · 2024-08-13
> >
> > Thank you to the authors for their rebuttal. I have reviewed all comments and the authors’ responses, which effectively addressed my concerns through clear explanations and additional experiments. I recommend incorporating these details into the manuscript or supplementary materials and support the acceptance of this paper.

---

> > > ### Author Response · Authors · 2024-08-13
> > > **Response to Reviewer byDA**
> > >
> > > Dear Reviewer byDA:
> > >
> > > Thank you for your recognition of our work. We appreciate your efforts in reviewing our work and rebuttal. We will incorporate your valuable suggestions into the revised version to ensure that it becomes a high-quality paper.
> > >
> > > Best Regards, Authors of 504.

---

### Official Review · Reviewer_aMJG · 2024-07-01

**Soundness:** 2
**Presentation:** 2
**Contribution:** 3
**Rating:** 4
**Confidence:** 5

**Summary:**

This paper presents a framework called TFGDA for semi-supervised graph domain adaptation (SGDA). It addresses the challenge of annotating unlabeled target graph nodes by utilizing knowledge from a source graph with limited labels. The framework incorporates three key strategies: Subgraph Topological Structure Alignment (STSA), Sphere-guided Domain Alignment (SDA), and Robustness-guided Node Clustering (RNC). These strategies collectively aim to encode topological structure information, stably align feature distributions, and guide the clustering of unlabeled nodes, thereby enhancing the model's transfer performance.

**Strengths:**

1. Interesting Topic: Graph domain adaptation (GDA) is a compelling topic with significant potential for advancing the field of graph-based learning.
2. SOTA Performance: The experimental results indicate that the proposed methods achieve state-of-the-art (SOTA) performance across various benchmarks.
3. Theoretical Foundation: The paper is supported by a solid theoretical foundation, leveraging topological data analysis and domain adaptation principles.

**Weaknesses:**

1. Limited Innovation: The innovation in the paper is limited. While the authors claim to be the first to consider graph structure, similar approaches have been explored in previous work that also utilize graph structural information.
2. Overly Optimistic Results: The experimental results seem excessively positive, with improvements of 5-10 points on certain datasets, which is highly challenging. The absence of open-source code makes it difficult to verify these results.
3. Lack of Code Availability: The paper does not provide open-source code, which hinders reproducibility and further validation of the proposed methods by the research community.

**Questions:**

please refer to Weaknesses

**Limitations:**

please refer to Weaknesses

---

> ### Author Rebuttal · Authors · 2024-08-07
>
> We extend our heartfelt gratitude to Reviewer aMJG for the careful reading of the manuscript and the valuable feedback provided, which are helpful to improve our paper. Our detailed point-by-point responses are provided below.
>
> **W1: While the authors claim to be the first to consider graph structure, similar approaches have been explored in previous work that also utilize graph structural information.**
> **A1:** Thanks for the valuable comment. We primarily showcase the innovation of our method through the following three aspects:
>
> **1):** Although many graph convolutional networks (GCNs)-based methods have already been proposed to exploit graph structure information to promote the learning of features, these methods heavily rely on a large amount of labeled data.
> Moreover, these methods overlook the substantial domain discrepancy between training and testing datasets in real-world scenarios, making them unsuitable for semi-supervised graph domain adaptation (SGDA) scenario.
> Specifically, these GCNs-based models trained only with the limited source labeled nodes (i.e., SGDA task scenario) will suffer severe performance degradation when directly applied to a new target domain.
>
> **2):** In addition, these GCNs-based methods [X1-X6] typically mine the graph structure information in the deep feature space by designing well-crafted GCN architectures or introducing  some complex modules. However, recent studies [X6-X8] have pointed out that GCNs are insufficient in capturing the sophisticated structure information in graph, which means that the graph structure information may be lost or destroyed after passing through the GCNs-based feature extractor. Thus, directly mining graph structure information from the deep feature space is a suboptimal way, which affects the learning of transferable node features in our SGDA setting.
>
> The proposed STSA strategy aims to extract the graph structure information directly from the input space and encode these powerful information into the latent spherical space by aligning the topological structures of the two spaces. This method does not lose or destroy the graph structure information during training.  Furthermore, our STSA strategy does not introduce any changes to the network architecture, effectively avoiding an increase in model's complexity and ensuring its adaptability to integration with other methods.
>
> **3):** What's more, these GCN-based methods typically model graph structure by considering the similarity between features of adjacent nodes, making it difficult to capture the sophisticated high-order structure information in graph [X5,X6].
> In contrast, our work attempts, for the first time, to model the graph structure information from a persistent homology (i.e., topological data analysis) perspective, enabling a more precise capture of the complex structures inherent in graph.
>
> In summary, our proposed STSA strategy is an innovative solution for mining graph structure information and facilitating graph transfer learning.
>
> Ref.[X1] Distilling Knowledge From Graph Convolutional Networks. CVPR 2020.\
> Ref.[X2] Graph-in-Graph Convolutional Network for Hyperspectral Image Classification. TNNLS 2022.\
> Ref.[X3] Multigraph Fusion for Dynamic Graph Convolutional Network. TNNLS 2022.\
> Ref.[X4] Knowledge Embedding Based Graph Convolutional Network. WWW 2021.\
> Ref.[X5] Scattering GCN: Overcoming Oversmoothness in Graph Convolutional Networks. NIPS 2020.\
> Ref.[X6] Coco: A Coupled Contrastive Framework for Unsupervised Domain Adaptive Graph Classification. ICML 2023.\
> Ref.[X7] Graph Kernel Neural Networks. TNNLS 2024.\
> Ref.[X8] Theoretically Improving Graph Neural Networks via Anonymous Walk Graph Kernels. WWW 2021.
>
> **W2&W3: The experimental results seem excessively positive. The paper does not provide open-source code.**
> **A2&A3:**\
> **1):** It is worth noting that semi-supervised graph domain adaptation (SGDA) is a relatively novel task scenario in graph transfer learning, first introduced in Ref.[X9]. Currently, there has been little exploration in this field. As stated in lines 248-251, the main compared methods adopted in Table 1 are some semi-supervised methods and graph domain adaptation methods.
> Due to the label scarcity of source graph and the huge domain discrepancy in the SGDA scenario, these semi-supervised methods are unable to tackle these challenges, resulting in severe performance degradation on the target domain.
>
> Although graph domain adaptation can alleviate domain discrepancy, they inevitably encounter the overfitting issue. Because these methods typically assume that the source domain is fully labeled, making it difficult to effectively utilize a large number of unlabeled nodes, as stated in lines 265-267.
> Our method effectively addresses these challenges, thus achieving SOTA performance.
>
> **2):** Regarding the pioneering work SGDA [X9], it struggles to achieve superior performance because it simply utilizes the classical adversarial learning to reduce domain discrepancy, and its clustering strategy inevitably introduces pseudo-label noise into the model (as verified in lines 307-308 and 518-532).
>
> Ref.[X9] Semi-supervised Domain Adaptation in Graph Transfer Learning. IJCAI 2023.
>
> **3) Source code:**
> Based on your valuable suggestion, we will provide AC with the source code of our method in order to validate the authority of our work. We commit to publicly release the code after the paper is accepted.
>
> If you have additional concerns, please let us know and we will do our best to address them. We appreciate your time and efforts in reviewing our work.

---

> ### Author Response · Authors · 2024-08-12
> **Looking forward to the reply**
>
> Dear Reviewer aMJG:
>
> Thanks so much again for the time and effort in reviewing our work. According to your valuable comments and concerns, we provide the source code and further discuss the related points. As the discussion period is nearing its end, we would like to kindly ask the reviewer if our rebuttal has addressed some of the reviewer's questions or concers and if any of them remain unaddressed. We would be happy to provide more details and clarifications.
>
> Best Regards, Authors of 504.

---

### Official Review · Reviewer_9hdx · 2024-07-13

**Soundness:** 3
**Presentation:** 3
**Contribution:** 3
**Rating:** 6
**Confidence:** 5

**Summary:**

This paper introduces a graph transfer learning framework called TFGDA, which leverages the structure information to enhance model’s generalization performance. The TFGDA framework includes the structure alignment strategy STSA, the feature distribution alignment strategy SDA, and the RNC strategy to address the overfitting caused by label scarcity. Experiments are conducted various benchmarks.

**Strengths:**

1.This paper is first attempt to utilize the intrinsic topological structure information hidden in graphs to assist graph transfer learning, which is well-motivated and novel.

2. This paper is well-written and well-organized. The figures are clear and the text is easy to follow.

3.The experiments are comprehensive. The experimental results demonstrate the superiority of the proposed framework.

**Weaknesses:**

1. This paper proposes to extract spherical features of subgraphs in Subgraph Topological Structure Alignment strategy. Please further explain the reasons for using spherical features and analyze the necessity.

2. The interplay between loss functions could affect the extracted features and model performance. It would be beneficial if the authors could provide a thorough analysis for the combination of loss functions and theoretically explanation of the success.

**Questions:**

See Weaknesses

**Limitations:**

The authors should further clarify the motivation behind using spherical features of their proposed method.

---

> ### Author Rebuttal · Authors · 2024-08-07
>
> We would like to express our gratitude to Reviewer 9hdx for their comprehensive evaluation of the manuscript and the insightful feedback provided, which will greatly contribute to the improvement of our paper. Kindly refer to our specific responses outlined below.
>
> **W1: This paper proposes to extract spherical features of subgraphs in STSA strategy. Please further explain the reasons for using spherical features and analyze the necessity. The authors should further clarify the motivation behind using spherical features of their proposed method.**
> **A1:** Thanks for your valuable advice.
>
> **1) explain the reasons:** Notably, several studies [Y1] in transfer learning have demonstrated that the domain shifts are mainly caused by the feature norm discrepancy between the source and target domains, and the model degradation on the target domain is primarily caused by its excessively smaller feature norms than the source domain. To eliminate the negative effects of feature norm differences, we map features into the spherical space to guide the alignment of feature distribution (as stated in lines 170-173), which can significantly facilitate the learning of domain-invariant features and improve the model's transfer performance.
>
> Furthermore, Refs.[Y1-Y2] have pointed out that highly informative features typically display more significant feature norms, and task-specific features with larger norms are more transferable.
> However, due to the large parameter search space, it is difficult to find the optimal spherical radius for different transfer tasks.
> To address this issue, we follow the lower bound theorem from Ref.[Y3] to set an appropriate radius $r$ for the spherical space $\mathbb{S}_{r}^{d-1}$, as mentioned in lines 173-176 of the manuscript.
>
> Ref.[Y1] Larger Norm More Transferable: An Adaptive Feature Norm Approach for
> Unsupervised Domain Adaptation. ICCV 2019. \
> Ref.[Y2] Rethinking the Smaller-Norm-Less-Informative Assumption in Channel Pruning of Convolution Layers. ICLR 2018.\
> Ref.[Y3] Unsupervised and Semi-supervised Robust Spherical Space Domain Adaptation. TPAMI 2022.
>
> **2) analyze the necessity**:
> According to your advice, we conduct additional ablation experiments to validate this point.  As described in Table 2 of the paper, variant TFGDA-T ($\mathcal{L}\_\{cls}+\mathcal{L}\_\{stsa}$) implements the STSA strategy in spherical space $\mathbb{S}\_\{r}^{d-1}$.
> Based on variant TFGDA-T, we remove the spherical space $\mathbb{S}\_\{r}^{d-1}$ and implement the STSA strategy in regular space, resulting in a new variant called TFGDA-T (w/o sphere).
> The experimental results are shown in the **Table A** of the global response PDF file.
> We can observe that the introduction of spherical space $\mathbb{S}\_\{r}^{d-1}$ can effectively enhance the model's transfer ability, which is consistent with the aforementioned analysis.
>
>
> **W2: The interplay between loss functions could affect the extracted features and model performance. It would be beneficial if the authors could provide a thorough analysis for the combination of loss functions and theoretically explanation of the success.**
> **A2:** Thanks for your valuable suggestion.
> It is worth noting that our ablation study results in Section 5.3 (lines 271-285) have validated the contribution of each loss term and the complementary relationship among different loss terms (i.e., different combinations of loss terms).  Moreover, the t-SNE visualization results in Figure 2 have also confirmed the importance the combination of loss terms $\mathcal{L}\_\{rnc}$, $\mathcal{L}\_\{stsa}$ and $\mathcal{L}\_\{sda}$.
> We have also conducted a sensitivity analysis on the trade-off parameters of these loss terms in Section A.3.2 of the Appendix (lines 557-565).
>
> **Theoretically Explanation:** Based on your insightful advice, we provide a theoretical analysis of our method. Due to the characters limit, we will present this theoretical analysis in the **global rebuttal area**.

---

> > ### Comment · Reviewer_9hdx · 2024-08-13
> >
> > Thanks for the authors' responses. I have adjusted my score accordingly, and vote for acceptance of this paper.

---

> > > ### Author Response · Authors · 2024-08-13
> > > **Response to Reviewer 9hdx**
> > >
> > > Dear Reviewer 9hdx:
> > >
> > > Thank you for the positive feedback. We appreciate your efforts in reviewing our work. We will reflect your suggestions in the revised version to enable it to be a high-quality paper.
> > >
> > > Best Regards, Authors of 504.

---

> ### Author Response · Authors · 2024-08-12
> **Looking forward to the reply**
>
> Dear Reviewer 9hdx:
>
> Thanks so much again for the time and effort in reviewing our work. As we are getting closer to the end of the author-reviewer discussion phase, we would like to kindly ask the reviewer if our rebuttal has addressed some of the reviewer's questions or concers and if any of them remain unaddressed. We would be happy to provide more details and clarifications.
>
> Best Regards, Authors of 504.

---

### Author Rebuttal · Authors · 2024-08-07

**1) To Reviewer 9hdx:** Dear Reviewer 9hdx, based on your insightful advice, we provide a **theoretical analysis** of our method.

The theoretical analysis of our method is based on the theory of domain adaptation (DA) [Y4-Y5].

Formally, let $\mathcal{H}$ be the hypothesis space. Given two domains $\mathcal{S}$ and $\mathcal{T}$, the probabilistic bound of error of hypothesis $h$ on the target domain is defined as:
$\\psi\_\{\\mathcal{T}}(h) \\le \\psi \_\{\\mathcal{S}}(h) + \\frac{1}{2}d\_\{\\mathcal{H}\\Delta \\mathcal{H}}(\\mathcal{S},\\mathcal{T})+\\mu^{\*\}$,
where the expected error on the target domain $\psi_{\mathcal{T}}(h)$ are bounded by three terms: **(1)** the expected error on source domain $\psi_{\mathcal{S}}(h)$; **(2)** the $\mathcal{H} \Delta \mathcal{H}$-divergence between the source and target domains $d_{\mathcal{H} \Delta \mathcal{H}}(\mathcal{S},\mathcal{T})$; **(3)** the combined error of ideal joint hypothesis $ \mu^{*} = \min_{h'\in \mathcal{H}} \psi_{\mathcal{S}}(h') + \psi_{\mathcal{T}}(h')$.

The goal of DA is to lower the upper bound of the expected target domain error $\psi_{\mathcal{T}}(h)$.
Note that in unsupervised domain adaptation (UDA), minimizing $\psi_{\mathcal{S}}(h)$ can be easily achieved with source label information, as source domain samples are completely annotated.
However, in our semi-supervised graph domain adaptation (SGDA) setting, due to the label scarcity of source domain, the model is prone to overfitting when solely relying on the source domain classification loss $\mathcal{L}\_\{cls}$ for optimization. Therefore, we introduce the **RNC** strategy ($\mathcal{L}\_\{rnc}$) to address the overfitting issue, with the aim of guiding $\psi\_\{\mathcal{S}}(h)$ towards further minimization.

Most DA methods mainly focus on reducing the domain discrepancy $d\_\{\mathcal{H} \Delta \mathcal{H}}(\mathcal{S},\mathcal{T})$, such as utilizing techniques like adversarial learning, MMD, and CORAL.
In comparison to these methods, our **SDA** strategy ($\mathcal{L}\_\{sda}$) effectively eliminates the feature norm discrepancy in spherical space $\mathbb{S}\_\{r}^{d-1}$ and guide a more stable alignment of feature distributions. Furthermore, considering that graph data contains rich structure information that encodes complex relationships among nodes and edges, and existing graph transfer learning (GTL) methods usually adopt graph convolutional networks (GCNs)-based feature extractor to learn domain-invariant node features. However, recent studies [Y6-Y8] have pointed out that GCNs are insufficient in capturing the sophisticated structure information in graph, which seriously affects the transfer of domain-invariant knowledge and consequently limits the model’s generalization ability. To solve this problem, we thus propose the **STSA** strategy ($\mathcal{L}\_\{cls}$) to align the topological structures of the input space and the spherical space, in order to facilitate the GCNs-based feature extractors to capture more domain-invariant node features.
Consequently, the combination of **SDA** ($\mathcal{L}\_\{sda}$) and **STSA** ($\mathcal{L}\_\{stsa}$) strategies further promotes the minimization of the domain discrepancy $d\_\{\mathcal{H} \Delta \mathcal{H}}(\mathcal{S},\mathcal{T})$.

Notably, $\\mu^{\*\}$ is expected to be extremely small, and therefore it is often neglected by previous methods.
However, it is possible that $\\mu^{\*\}$ tends to be large when the cross-domain category distributions are not well aligned [Y9].
In this paper, we leverage the **RNC** strategy ($\mathcal{L}\_\{rnc}$) to guide both labeled nodes and unlabeled nodes toward achieving robust clustering, effectively promoting the fine-grained alignment of category distributions and  ensuring that $\\mu^{\*\}$ remains at a relatively small value.

In summary, our proposed method not only minimizes the expected error on source domain $\psi\_\{\mathcal{S}}(h)$ and the domain discrepancy $d\_\{\mathcal{H} \Delta \mathcal{H}}(\mathcal{S},\mathcal{T})$, but also keeps $\\mu^{\*\}$ at a small value, thereby ensuring a low upper bound.

Ref.[Y4] A theory of learning from different domains. Machine learning. 2010.\
Ref.[Y5] Analysis of representations for domain adaptation. NIPS 2006.\
Ref.[Y6] Coco: A coupled contrastive framework for unsupervised domain adaptive graph classification. ICML 2023.\
Ref.[Y7] Graph Kernel Neural Networks. TNNLS 2024.\
Ref.[Y8] Theoretically improving graph neural networks via anonymous walk graph kernels. WWW 2021.\
Ref.[Y9] Progressive feature alignment for unsupervised domain adaptation. CVPR 2019.

**2) To Reviewer 9hdx:** The necessity analysis Table A mentioned in A1 is displayed in the global response PDF file.

**3) To All Reviewers:** Based on the valuable suggestion of Reviewer aMJG, we will provide AC with the source code of our method in order to validate the authority of our work. You can get the anonymous link of our source code from the AC.

---

> ### Author Response · Authors · 2024-08-07
> **Anonymous link to the source code**
>
> Dear AC and all Reviewers:
>
> Based on the valuable suggestion of Reviewer **aMJG**, we provide the source code and pre-trained models of our method to demonstrate the authority of our work. **Code and pre-trained models are available at:** https://anonymous.4open.science/r/TFGDA-003D
>
> We appreciate your time and efforts in reviewing our work.
>
> Best Regards, Authors of 504.

---

### Comment · Area_Chair_3qny · 2024-08-09
**Discussion period starts**

Dear reviewers,

Thank you for your valuable contributions to the NeurIPS review process! The author-reviewer discussion period has now begun. I’ve noticed that the ratings for this paper are quite dispersed, and opinions among the reviewers are not fully aligned. This makes our discussion even more crucial to reach a consensus and ensure a fair and thorough evaluation. Please engage actively with the authors during this period. If you have any questions or need further clarification on any points, this is the best time to address them directly with the authors.


best,

AC

---

> ### Comment · Reviewer_R5oT · 2024-08-11
> **my concerns have been well addressed**
>
> Thank you to the authors for their detailed response to my questions. After reviewing all the comments and the rebuttal materials, my concerns have been well addressed. This process has further deepened my appreciation for the design and reproducibility of the proposed approach. I have no further questions. I am happy to adjust my rating to reflect my positive feedback.

---

### Decision · Program_Chairs · 2024-09-25

**Decision:**

Accept (poster)

**Comment:**

**Summary**

This paper presents TFGDA, which addresses the challenges in semi-supervised graph domain adaptation. The reviewers provided a generally positive assessment of the paper, noting its novelty, clear motivation, and thorough experimental validation. The proposed methods were recognized for their ability to effectively utilize graph structure information to improve transfer performance, which is a contribution to the field of graph transfer learning. The extensive experiments demonstrated the superiority of TFGDA over existing state-of-the-art methods.

**Decision**

The decision to accept is based on the novelty of leveraging the topological structures of graphs of graph structure information for GDA.
In the final version, the authors should incorporate the additional experiments conducted during the rebuttal period, particularly those involving the AliCVR dataset, and clarify the distinctions between your approach and existing works. Additionally, ensure that the practical applications of your method, as well as a summary of the SDA strategy in an algorithm table, are clearly presented. These enhancements will further strengthen the paper's contribution and clarity.